# Mildly Overparameterized ReLU Networks Have a Favorable Loss Landscape

## Abstract

We study the loss landscape of two-layer mildly overparameterized ReLU neural networks on a generic finite input dataset for the squared error loss. Our approach involves bounding the dimension of the sets of local and global minima using the rank of the Jacobian of the parameterization map. Using results on random binary matrices, we show most activation patterns correspond to parameter regions with no bad differentiable local minima. Furthermore, for one-dimensional input data, we show most activation regions realizable by the network contain a high dimensional set of global minima and no bad local minima. We experimentally confirm these results by finding a phase transition from most regions having full rank to many regions having deficient rank depending on the amount of overparameterization.

## 1 Introduction

The optimization landscape of neural networks has been a topic of enormous interest over the years. A particularly puzzling question is why bad local minima do not seem to be a problem for training. In this context, an important observation is that overparameterization can yield a more benevolent optimization landscapes. While this type of observation can be traced back at least to the works of Poston et al. (1991); Gori & Tesi (1992), it has increasingly become a focus of investigation in recent years. In this article we follow this line of work and describe the optimization landscape of a moderately overparameterized shallow ReLU network with a view on the different activation regions of the parameter space. Before going into the details of our results, we first provide some context.

The existence of non-global local minima has been documented in numerous works. This is the case even for networks without hidden units (Sontag & Sussmann, 1989) and single units (Auer et al., 1995). For shallow networks, Fukumizu & Amari (2000) showed how local minima and plateaus can arise from the hierarchical structure of the model for general types of targets, loss functions and activation functions. Other works have also constructed concrete examples of non-global local minima for several architectures (Swirszcz et al., 2017). While these works considered finite datasets, Safran & Shamir (2018) observed that spurious minima are also common for the student-teacher population loss of a two-layer ReLU network with unit output weights. In fact, for ReLU networks Yun et al. (2019b); Goldblum et al. (2020) show that if linear models cannot perfectly fit the data, one can construct local minima that are not global. He et al. (2020) shows the existence of spurious minima for arbitrary piecewise linear (non-linear) activations and arbitrary continuously differentiable loss functions.

A number of works suggest overparameterized networks have a more benevolent optimization landscape than underparameterized ones. For instance, Soudry & Carmon (2016) show for mildly overparameterized networks with leaky ReLUs that for almost every dataset every differentiable local minimum of the squared error training loss is a zero-loss global minimum, provided a certain type of dropout noise is used. In addition, for the case of one hidden layer they show this is the case whenever the number of weights in the first layer matches the number of training samples, $d_0 d_1 \geq n$. In another work, Safran & Shamir (2016) used a computer aided approach to study the existence of spurious local minima, arriving at the key finding that when the student and teacher networks are equal in width spurious local minima are commonly encountered. However, under only mild overparameterization, no spurious local minima were identified, implying that overparameterization leads to a more benign loss landscape. In the work of Tian (2017), the student and teacher networks are assumed to be equal in width and it is shown that if the dimension of the data is sufficiently

large relative to the width, then the critical points lying outside the span of the ground truth weight vectors of the teacher network form manifolds. Another related line of works studies the connectivity of sublevel sets (Nguyen, 2019) and bounds on the overparameterization that ensures existence of descent paths (Sharifnassab et al., 2020).

The key objects in our investigation will be the rank of the Jacobian of the parametrization map over a finite input dataset, and the combinatorics of the corresponding subdivision of parameter space into pieces where the loss is differentiable. The Jacobian map captures the local dependency of the functions represented over the dataset on their parameters. The notion of connecting parameters and functions is prevalent in old and new studies of neural networks: for instance, in discussions of parameter symmetries (Chen et al., 1993), functional equivalence (Phuong & Lampert, 2020), functional dimension (Grigsby et al., 2022), the question of when a ReLU network is a universal approximator over a finite input dataset (Yun et al., 2019a), as well as in studies involving the neural tangent kernel (Jacot et al., 2018).

We highlight a few works that take a geometric or combinatorial perspective to discuss the optimization landscape. Using dimension counting arguments, Cooper (2021) showed that, under suitable overparameterization and smoothness assumptions, the set of zero-training loss parameters has the expected codimension, equal to the number of training data points. In the case of ReLU networks, Laurent & von Brecht (2018) used the piecewise multilinear structure of the parameterization map to describe the location of the minimizers of the hinge loss. Further, for piecewise linear activation functions Zhou & Liang (2018) partition the parameter space into cones corresponding to different activation patterns of the network to show that, while linear networks have no spurious minima, shallow ReLU networks do. In a similar vein, Liu (2021) study one hidden layer ReLU networks and show for a convex loss that differentiable local minima in an activation region are global minima within the region, as well as providing conditions for the existence of differentiable local minima, saddles and non differentiable local minima. Considering the parameter symmetries, Simsek et al. (2021) show that the level of overparameterization changes the relative number of subspaces that make up the set of global minima. As a result, overparameterization implies a larger set of global minima relative to the set of critical points, and underparameterization the reverse. Wang et al. (2022) studied the optimization landscape in two-layer ReLU networks and showed that all optimal solutions of the non-convex loss can be found via the optimal solutions of a convex program.

## 1.1 CONTRIBUTIONS

We show for two-layer ReLU neural networks that most linear regions of parameter space have no bad differentiable local minima and often contain a high-dimensional space of global minima. We examine the loss landscape for various scalings of the input dimension $d_0$, hidden dimension $d_1$, and number of data points $n$.

- Theorem 5 shows that if $d_0 d_1 \geq n$ and $d_1 = \Omega(\log(\frac{n}{\epsilon d_0}))$, then all activation regions, except for a fraction of size at most $\epsilon$, have no bad local minima. We establish this by studying the rank of the Jacobian with respect to the parameters. By appealing to results from random matrix theory on binary matrices, we show for most choices of activation patterns that the Jacobian will have full rank. Given this, all local minima within a region will be zero-loss global minima. For generic high-dimensional input data $d_0 \geq n$, this implies most *non-empty* activation regions will have no bad local minima as all activation regions are non-empty for generic data in high dimensions.

- In Theorem 10, we specialize to the case of one-dimensional input data $d_0 = 1$, and consider two-layer networks with a bias term. We show that if $d_1 = \Omega(n \log(\frac{n}{\epsilon}))$, all but at most a fraction $\epsilon$ of *non-empty* linear regions in parameter space will have no bad local minima. In contrast to Theorem 5 which looks at all binary matrices of potential activation patterns, in the one-dimensional case we are able to explicitly enumerate the binary matrices which correspond to non-empty activation regions.

- Theorem 12 continues our investigation of one-dimensional input data, this time concerning the existence of global minima within a region. Suppose that the output head $v$ has $d_+$ positive weights and $d_-$ negative weights and $d_+ + d_- = d_1$. We show that if $d_+, d_- = \Omega(n \log(\frac{n}{\epsilon}))$, then all but at most a fraction $\epsilon$ of non-empty linear regions in parameter space will have global minima. Moreover, the regions with global minima will contain an affine set of global minima of codimension $n$.

## 1.2 RELATION TO PRIOR WORKS

As indicated above, several works have studied sets of local and global critical points of two-layer ReLU networks, so it is in order to give a comparison. We take a different approach by identifying the *number of regions* which have a favorable optimization landscape, and as a result are able to avoid having to make certain assumptions about the dataset or network. Since we are able to exclude pathological regions with many dead neurons, we can formulate our results for ReLU activations rather than LeakyReLU or other smooth activation functions. In contrast to Soudry & Carmon (2016), we do not assume dropout noise on the outputs of the network, and as a result global minima in our setting typically attain zero loss. Unlike Safran & Shamir (2018), we do not assume any particular distribution on our datasets; our results hold for almost all datasets (a set of full Lebesgue measure). Extremely overparameterized networks (with $d_1 = \Omega(n^2)$) are known to follow lazy training (see Chizat et al., 2019); our theorems hold under more realistic assumptions of mild overparameterization $d_1 = \Omega(n \log n)$ or even $d_1 = \Omega(1)$ for high-dimensional inputs. We are able to avoid excessive overparameterization by emphasizing *qualitative* aspects of the loss landscape, using only the rank of the Jacobian rather than the smallest eigenvalue of the neural tangent kernel, for instance.

## 2 PRELIMINARIES

Before specializing to specific neural network architectures, we introduce general definitions which encompass all of the models we will study. For any $d \in \mathbb{N}$ we will write $[d] := \{1, \ldots, d\}$. We will write $\mathbf{1}_d$ for a vector of $d$ ones, and drop the subscript when the dimension is clear from the context. The Hadamard (entry-wise) product of two same-format matrices $A$ and $B$ is defined as $A \odot B := (A_{ij} \cdot B_{ij})$. The Kronecker product of two vectors $u \in \mathbb{R}^n$ and $v \in \mathbb{R}^m$ is defined as $u \otimes v := (u_i v_j) \in \mathbb{R}^{nm}$.

Let $\mathbb{R}[x_1, \ldots, x_n]$ denote the set of polynomials in variables $x_1, \ldots, x_n$ with real coefficients. We say that a set $\mathcal{V} \subseteq \mathbb{R}^n$ is an *algebraic set* if there exist $f_1, \ldots, f_m \in \mathbb{R}[x_1, \ldots, x_n]$ such that $\mathcal{V}$ is the zero locus of the polynomials $f_i$; that is,

$$\mathcal{V} = \{x \in \mathbb{R}^n : f_i(x) = 0 \text{ for all } i \in [m]\}.$$

Clearly, $\emptyset$ and $\mathbb{R}^n$ are algebraic, being zero sets of $f = 1$ and $f = 0$ respectively. A finite union of algebraic subsets is algebraic, as well as an arbitrary intersection of algebraic subsets. In other words, algebraic sets form a *topological space*. Its topology is known as the *Zariski topology*. The following lemma is a basic fact of algebraic geometry, and follows from subdividing algebraic sets into submanifolds of $\mathbb{R}^n$.

**Lemma 1.** *Let $\mathcal{V}$ be a proper algebraic subset of $\mathbb{R}^n$. Then $\mathcal{V}$ has Lebesgue measure $0$.*

For more details on the above facts, we refer the reader to Harris (2013). Justified by Lemma 1, we say that a property $P$ depending on $x \in \mathbb{R}^n$ holds *for generic $x$* if there exists a proper algebraic set $\mathcal{V}$ such that $P$ holds whenever $x \notin \mathcal{V}$. So if $P$ is a property that holds for generic $x$, then in particular it holds for a set of full Lebesgue measure.

We consider input data

$$X = (x^{(1)}, \ldots, x^{(n)}) \in \mathbb{R}^{d \times n}$$

and output data

$$y = (y^{(1)}, \ldots, y^{(n)}) \in \mathbb{R}^{1 \times n}.$$

A *parameterized model* with parameter space $\mathbb{R}^m$ is a mapping

$$F : \mathbb{R}^m \times \mathbb{R}^d \to \mathbb{R}.$$

We overload notation and also define $F$ as a map from $\mathbb{R}^m \times \mathbb{R}^{d \times n}$ to $\mathbb{R}^{1 \times n}$ by

$$F(\theta, (x^{(1)}, \ldots, x^{(n)})) := (F(\theta, x^{(1)}), F(\theta, x^{(2)}), \ldots, F(\theta, x^{(n)})).$$

Whether we are thinking of $F$ as a mapping on individual data points or on datasets will be clear from the context. We define the mean squared error loss $L : \mathbb{R}^m \times \mathbb{R}^{d \times n} \times \mathbb{R}^{1 \times n} \to \mathbb{R}^1$ by

$$L(\theta, X, y) := \frac{1}{2} \sum_{i=1}^{n} (F(\theta, x^{(i)}) - y^{(i)})^2$$

$$= \frac{1}{2} \|F(\theta, X) - y\|^2. \tag{1}$$

For a fixed dataset $(X, y)$, let $\mathcal{G}_{X,y} \subseteq \mathbb{R}^m$ denote the set of global minima of the loss $L$; that is,

$$\mathcal{G}_{X,y} = \left\{ \theta \in \mathbb{R}^m : L(\theta, X, y) = \inf_{\phi \in \mathbb{R}^m} L(\phi, X, y) \right\}.$$

If there exists $\theta^* \in \mathbb{R}^m$ such that $F(\theta^*, X) = y$, then $L(\theta^*, X, y) = 0$, so $\theta^*$ is a global minimum. In such a case,

$$\mathcal{G}_{X,y} = \{\theta \in \mathbb{R}^m : F(\theta, X) = y\}.$$

For a dataset $(X, y)$, we say that $\theta \in \mathbb{R}^m$ is a *local minimum* if there exists an open set $\mathcal{U} \subseteq \mathbb{R}^m$ containing $\theta$ such that $L(\theta, X, y) \leq L(\phi, X, y)$ for all $\phi \in \mathcal{U}$. We say that $\theta$ is a *bad local minimum* if it is a local minimum but not a global minimum.

A key indicator of the local optimization landscape of a neural network is the rank of the Jacobian of the map $F$ with respect to the parameters $\theta$. We will use the following observation.

**Lemma 2.** *Fix a dataset $(X, y) \in \mathbb{R}^{d \times n} \times \mathbb{R}^{1 \times n}$. Let $F$ be a parameterized model and let $\theta \in \mathbb{R}^m$ be a differentiable critical point of the squared error loss equation 1. If $\mathrm{rank}(\nabla_\theta F(\theta, X)) = n$, then $\theta$ is a global minimizer.*

*Proof.* Suppose that $\theta \in \mathbb{R}^m$ is a differentiable critical point of $L$. Then

$$\begin{aligned} 0 &= \nabla_\theta L(\theta, X, y) \\ &= \nabla_\theta F(\theta, X) \cdot (F(\theta, X) - y). \end{aligned}$$

Since $\mathrm{rank}(\nabla_\theta F(\theta, X, y)) = n$, this implies that $F(\theta, X) - y = 0$. In other words, $\theta$ is a global minimizer. $\qquad\square$

## 3 COUNTING ACTIVATION REGIONS WITH NO BAD LOCAL MINIMA

We now focus on a two-layer network with $d_0$ inputs, one hidden layer of $d_1$ ReLUs, and an output layer. Our parameter space is the input weight matrix in $\mathbb{R}^{d_1 \times d_0}$. Our input dataset will be an element $X \in \mathbb{R}^{d_0 \times n}$ and our output dataset an element $y \in \mathbb{R}^{1 \times n}$, where $n$ is the cardinality of the dataset. The model $F : (\mathbb{R}^{d_1 \times d_0} \times \mathbb{R}^{d_1}) \times \mathbb{R}^{d_0} \to \mathbb{R}$ is defined by

$$F(W, v, x) = v^T \sigma(Wx),$$

where $\sigma$ is the ReLU activation function $s \mapsto \max\{0, s\}$ applied componentwise. We write $W = (w^{(1)}, \ldots, w^{(d_1)})^T$, where $w^{(i)} \in \mathbb{R}^{d_0}$ is the $i$th row of $W$. Since $\sigma$ is piecewise linear, for any finite input dataset $X$ we may split the parameter space into a finite number of regions on which $F$ is linear in $W$ (and linear in $v$). For any binary matrix $A \in \{0, 1\}^{d_1 \times n}$ and input dataset $X \in \mathbb{R}^{d_0 \times n}$, we define a corresponding *activation region* in parameter space by

$$\mathcal{S}_X^A := \left\{ W \in \mathbb{R}^{d_1 \times d_0} : (2A_{ij} - 1)\langle w^{(i)}, x^{(j)} \rangle > 0 \text{ for all } i \in [d_1], j \in [n] \right\}.$$

This is a polyhedral cone defined by linear inequalities for each $w^{(i)}$. For each $A \in \{0, 1\}^{d_1 \times n}$ and $X \in \mathbb{R}^{d_0 \times n}$, we have $F(W, v, X) = v^T A \odot (WX)$ for all $W \in \mathcal{S}_X^A$, which is linear in $W$. The Jacobian with respect to the parameters is

$$\nabla_\theta F(W, X) = [v_i A_{ij} x^{(j)}]_{ij} = [(v \odot a^{(j)}) \otimes x^{(j)}]_j, \quad \text{for all } W \in \mathcal{S}_X^A, A \in \{0, 1\}^{d_1 \times n},$$

where $a^{(j)}$ denotes the $j$th column of $A$. In order to establish nonsingularity of the Jacobian $\nabla_\theta F$, we need to ensure that the activation matrix $A$ does not have too much linear dependence between its rows. The following result, due to Bourgain et al. (2010), establishes this for most choices of $A$.

**Theorem 3.** *Let $A$ be a $d \times d$ matrix whose entries are iid random variables sampled uniformly from $\{0, 1\}$. Then $A$ is singular with probability at most*

$$\left( \frac{1}{\sqrt{2}} + o(1) \right)^d.$$

The next lemma shows that the specific values of $v$ are not relevant to the rank of the Jacobian.

**Lemma 4.** *Let $a^{(j)} \in \mathbb{R}^{d_1}$, $x^{(j)} \in \mathbb{R}^{d_0}$ for $j \in [n]$ and $v \in \mathbb{R}^{d_1}$ be vectors, with $v_i \neq 0$ for all $i \in [d_1]$. Then*

$$\text{rank}(\{(v \odot a^{(j)}) \otimes x^{(j)} : j \in n\}) = \text{rank}(\{a^{(j)} \otimes x^{(j)} : j \in [n]\}).$$

Using algebraic techniques, we show that for generic $X$, the rank of the Jacobian is determined by $A$. Then by the above results, Lemma 2 concludes that for most activation regions the smooth critical points are global minima (see full details in Appendix A):

**Theorem 5.** *Let $\epsilon > 0$. If*

$$d_1 \geq \max\left(\frac{n}{d_0}, \Omega\left(\log\left(\frac{n}{\epsilon d_0}\right)\right)\right),$$

*then for generic datasets $(X, y)$, the following holds. In all but at most $\lceil \epsilon 2^{nd_1} \rceil$ activation regions (i.e., an $\epsilon$ fraction of all regions), every differentiable critical point of $L$ with nonzero entries for $v$ is a global minimum.*

## 4 COUNTING NON-EMPTY ACTIVATION REGIONS

In this section, we evaluate the number of non-empty activation regions in parameter space for a given input dataset $X$. The regions in parameter space are separated by the hyperplanes $\{W : \langle w^{(i)}, x^{(j)} \rangle = 0\}$, $i \in [d_1]$, $j \in [n]$. Standard results on hyperplane arrangements give the following proposition. We say that a set of vectors in a $d$-dimensional space is in general position if any $k \leq d$ of them are linearly independent, which is a generic property.

**Proposition 6** (Number of non-empty regions). *Consider a network with one layer of $d_1$ ReLUs. If the columns of $X$ are in general position in a $d$-dimensional linear space, then the number of non-empty activation regions in the parameter space is $(2 \sum_{k=0}^{d-1} \binom{n-1}{k})^{d_1}$.*

The formula in the above statement is equal to $2^{nd_1}$ if and only if $n \leq d$ and $O(n^{dd_1})$ if $n > d$. For an arbitrary dataset that is not necessarily in general position, the regions can be enumerated using a celebrated formula by Zaslavsky (1975), in terms of the intersection poset of the hyperplanes. Thus we have a relatively good understanding of *how many* activation regions are non-empty. One can show that the maximal number of non-empty regions is attained when the dataset is in general position. Notably, the number is the same for any dataset that is in general position.

On the other hand, the *identity* of the non-empty regions depends more closely on the specific dataset and is harder to catalogue. For a given dataset $X$ the non-empty regions correspond to the vertices of the Minkowski sum of the line segments with end points $0, x^{(j)}$, for $j \in [n]$, as can be inferred from results in tropical geometry (see Joswig, 2021).

**Proposition 7** (Identity of non-empty regions). *Let $A \in \{0, 1\}^{d_1 \times n}$. The corresponding activation region is non-empty if and only if $\sum_{j:A_{ij}=1} x^{(j)}$ is a vertex of $\sum_{j \in [n]} \text{conv}\{0, x^{(j)}\}$ for all $i \in [d_1]$.*

This provides a sense of which activation regions are non-empty, depending on $X$. The explicit list of non-empty regions is known in the literature as an oriented matroid (see Björner et al., 1999), a sort of combinatorial type of the dataset.

We observe that if $d$ is large in relation to $n$ and the data is in general position, then by Proposition 6 most activation regions are non empty. Thus we obtain the following corollary of Theorem 5.

**Corollary 8.** *Under the same assumptions as Theorem 5, if $d \geq n$, then for $X$ in general position and arbitrary $y$, the following holds. In all but at most an $\epsilon$ fraction of all non-empty activation regions, every differentiable critical point of $L$ with nonzero entries for $v$ is a zero loss global minimum.*

## 5 COUNTING NON-EMPTY ACTIVATION REGIONS WITH NO BAD LOCAL MINIMA

We now take a closer look at case of a single input dimension, $d_0 = 1$. Consider a two-layer ReLU network with input dimension $d_0 = 1$, hidden dimension $d_1$, and a dataset consisting of $n$ data points.

We suppose the network has a bias term $b \in \mathbb{R}^{d_1}$. So the model $F : (\mathbb{R}^{d_1} \times \mathbb{R}^{d_1} \times \mathbb{R}^{d_1}) \times \mathbb{R} \to \mathbb{R}$ is given by

$$F(w, b, v, x) = v^T \sigma(wx + b).$$

Since we include a bias term here, we define the activation region $\mathcal{S}_X^A$ by

$$\mathcal{S}_X^A := \left\{ (w, b) \in \mathbb{R}^{d_1} \times \mathbb{R}^{d_1} : (2A_{ij} - 1)(w^{(i)}x^{(j)} + b^{(i)}) > 0 \text{ for all } i \in [d_1], j \in [n] \right\}.$$

In this one-dimensional case, we obtain new bounds on the fraction of favorable activation regions to show that most *non-empty* activation regions have no bad differentiable local minima. We begin with a characterization of which activation regions are non-empty. For $k \in [n+1]$, we introduce the *step vectors* $\xi^{(k,0)}, \xi^{(k,1)} \in \mathbb{R}^n$, defined by

$$(\xi^{(k,0)})_i = \begin{cases} 1 & \text{if } i < k, \\ 0 & \text{if } i \geq k \end{cases}, \quad \text{and} \quad (\xi^{(k,1)})_i = \begin{cases} 0 & \text{if } i < k, \\ 1 & \text{if } i \geq k \end{cases}.$$

Note that $\xi^{(1,0)} = \xi^{(n+1,1)} = 0$ and $\xi^{(n+1,0)} = \xi^{(1,1)} = 1$. There are $2n$ step vectors in total. Intuitively, step vectors describe activation regions because all data points on one side of a threshold value activate the neuron. The following lemma makes this notion precise.

**Lemma 9.** *Fix a dataset $(X, y)$ with $x^{(1)} < x^{(2)} < \cdots < x^{(n)}$. Let $A \in \{0, 1\}^{d_1 \times n}$ be a binary matrix. Then $\mathcal{S}_X^A$ is non-empty if and only if the rows of $A$ are step vectors. In particular, there are exactly $(2n)^{d_1}$ non-empty activation regions.*

Using this characterization of the non-empty activation regions, we show that most activation patterns corresponding to these regions yield full-rank Jacobian matrices, and hence the regions have no bad local minima.

**Theorem 10.** *Let $\epsilon \in (0, 1)$. Suppose that $X$ consists of distinct data points, and*

$$d_1 \geq 2n \log\left(\frac{n}{\epsilon}\right).$$

*Then in all but at most a fraction $\epsilon$ of non-empty activation regions, $\nabla_\theta F$ is full rank and every differentiable critical point of $L$ where $v$ has nonzero entries is a global minimum.*

Our strategy for proving Theorem 10 hinges on the following observation. For the sake of example, consider the step vectors $\xi^{(1,1)}, \xi^{(2,1)}, \cdots, \xi^{(n,1)}$. This set of vectors forms a basis of $\mathbb{R}^n$, so if each of these vectors was a row of the activation matrix $A$, it would have full rank. This observation generalizes to cases where some of the step vectors are taken to be $\xi^{(k,0)}$ instead of $\xi^{(k,1)}$. If enough step vectors are "collected" by rows of the activation matrix, it will be of full rank. We can interpret this condition in a probabilistic way. Suppose that the rows of $A$ are sampled randomly from the set of step vectors. We wish to determine the probability that after $d_1$ samples, enough step vectors have been sampled to cover a certain set. We use the following lemma.

**Lemma 11** (Coupon collector's problem). *Let $\epsilon \in (0, 1)$, and let $n \leq m$ be positive integers. Let $C_1, C_2, \ldots, C_d \in [m]$ be iid random variables such that for all $j \in [n]$ one has $\mathbb{P}(C_1 = j) \geq \delta$. If*

$$d \geq \frac{1}{\delta} \log\left(\frac{n}{\epsilon}\right),$$

*then $[n] \subseteq \{C_1, \ldots, C_d\}$ with probability at least $1 - \epsilon$.*

This gives us a bound for the probability that a randomly sampled region is of full rank. We finally convert this into a combinatorial statement to obtain Theorem 10. For the details and a complete proof, see Appendix C.

In one-dimensional input space, the existence of global minima within a region requires similar conditions to the region having a full rank Jacobian. Both of them depend on having many different step vectors represented among the rows of the activation matrix. The condition we need to check for the existence of global minima is slightly more stringent, and depends on there being enough step vectors for both the positive and negative entries of $v$.

**Theorem 12** (Fraction of regions with global minima). *Let $\epsilon \in (0,1)$ and let $v \in \mathbb{R}_1^d$ have nonzero entries. Suppose that $X$ consists of distinct data points,*

$$|\{i \in [d_1] : v^{(i)} > 0\}| \geq 2n \log\left(\frac{2n}{\epsilon}\right),$$

*and*

$$|\{i \in [d_1] : v^{(i)} < 0\}| \geq 2n \log\left(\frac{2n}{\epsilon}\right).$$

*Then in all but at most an $\epsilon$ fraction of non-empty activation regions $\mathcal{S}_X^A$, the subset of global minimizers in $(w, b)$, $\mathcal{G}_{X,y} \cap \mathcal{S}_X^A$, is a non-empty affine set of codimension $n$. Moreover, all global minima of $L$ have zero loss.*

We provide the proof of this statement in Appendix C.

We have shown in both the one-dimensional and higher-dimensional cases that most activation regions have a full rank Jacobian and contain no bad local minima. This suggests that a large fraction of parameter space by volume should also have a full rank Jacobian. This is indeed the case, and we prove this in Appendix F using the same techniques as our main results.

## 6 FUNCTION SPACE ON ONE-DIMENSIONAL DATA

We have studied activation regions in parameter space over which the Jacobian has full rank. We can give a picture of what the function space looks like as follows. We describe the function space of a ReLU over the data and based on this the function space of a network with a hidden layer of ReLUs.

Consider a single ReLU with one input with bias on $n$ input data points in $\mathbb{R}$. Equivalently, this is as a ReLU with two input dimensions and no bias on $n$ input data points in $1 \times \mathbb{R}$. For fixed $X$, denote by $F_X = [f(x^{(1)}), \ldots, f(x^{(n)})]$ the vector of outputs on all input data points, and by $\mathcal{F}_X = \{F_X(\theta) : \theta\}$ the set of all such vectors for any choice of the parameters. For a network with a single output coordinate, this is a subset of $\mathbb{R}^X$. As before, without loss of generality we sort the input data as $x^{(1)} < \cdots < x^{(n)}$ (according to the non-constant component). We will use notation of the form $X_{\geq i} = [0, \ldots, 0, x^{(i)}, \ldots, x^{(n)}]$. Further, we write $\bar{x}^{(i)} = [x_2^{(i)}, -1]$, which is a solution of the linear equation $\langle w, x^{(i)} \rangle = 0$. Recall that a polyline is a list of points with line segments drawn between consecutive points.

Since the parametrization map is piecewise linear, the function space is the union of the images of the Jacobian over the parameter regions where it is constant. In the case of a single ReLU one quickly verifies that, for $n = 1$, $\mathcal{F}_X = \mathbb{R}_{\geq 0}$, and for $n = 2$, $\mathcal{F}_X = \mathbb{R}_{\geq 0}^2$. For general $n$, there will be equality and inequality constraints, coming from the bounded rank of the Jacobian and the boundaries of the linear regions in parameter space. In Appendix D we offer details for the following statement.

**Proposition 13** (Function space on one-dimensional data). *Let $X$ be a list of $n$ distinct points in $1 \times \mathbb{R}$ sorted in increasing order with respect to the second coordinate.*

- *Then the set of functions a ReLU represents on $X$ is a polyhedral cone consisting of functions $\alpha f$, where $\alpha \geq 0$ and $f$ is an element of the polyline with vertices*

$$\bar{x}^{(i)} X_{\leq i}, \ i = 1, \ldots, n \quad and \quad -\bar{x}^{(i)} X_{\geq i}, \ i = 1, \ldots, n. \tag{2}$$

- *The set of functions represented by a sum of $m$ ReLUs consists of non-negative scalar multiples of convex combinations of any $m$ points on this polyline.*

- *The set of functions represented by arbitrary linear combinations of $m$ ReLUs consists of arbitrary scalar multiples of affine combinations of any $m$ points on this polyline.*

The function space of a single ReLU on $3$ and $4$ data points is illustrated in Figure 1.

For a single ReLU, there are $2n$ non-empty activation regions in parameter space. One of them has Jacobian rank 0 and is mapped to the 0 function, two others have Jacobian rank 1 and are mapped to non-negative scalar multiples of the coordinate vectors $e_1$ and $e_n$, and the other $2n - 3$ regions have

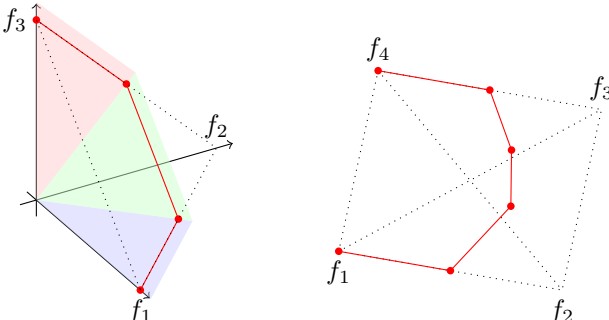

Figure 1: Function space of a ReLU on $n$ data points in $\mathbb{R}$, for $n = 3, 4$. The function space is a polyhedral cone in the non-negative orthant of $\mathbb{R}^n$. We can represent this, up to scaling by non-negative factors, by functions $f = (f_1, \ldots, f_n)$ with $f_1 + \cdots + f_n = 1$. These form a polyline, shown in red, inside the $(n-1)$-simplex. The sum of $m$ ReLUs corresponds to non-negative multiples of convex combinations of any $m$ points in the polyline, and arbitrary linear combinations of $m$ ReLUs correspond to arbitrary scalar multiples of affine combinations of any $m$ points in this polyline.

Jacobian rank 2 and are each mapped to the set of non-negative scalar multiples of a line segment in the polyline. Vertices in the list equation 2 correspond to the extreme rays of the linear pieces of the function space of the ReLU. They are the extreme rays of the cone of non-negative convex functions on $X$. Here a function on $X$ is convex if $\frac{f(x^{(j+1)}) - f(x^{(i)})}{x^{(i+1)} - x^{(i)}} \leq \frac{f(x^{(i+2)}) - f(x^{(i+1)})}{x^{(i+2)} - x^{(i+1)}}$. A non-negative sum of ReLUs is always contained in the cone of non-negative convex functions, which over $n$ data points is an $n$-dimensional convex polyhedral cone. For an overparameterized network, if an activation region in parameter space has full rank Jacobian, then that region maps to an $n$-dimensional polyhedral cone. It contains a zero-loss minimizer if and only if the corresponding cone intersects the output data vector $y \in \mathbb{R}^X$.

## 7 EXPERIMENTS

We empirically demonstrate that most regions of parameter space have a good optimization landscape by computing the rank of the Jacobian for two-layer neural networks. We initialize our network with random weights and biases sampled iid uniformly on $\left[-\frac{1}{\sqrt{d_1}}, \frac{1}{\sqrt{d_1}}\right]$. We evaluate the network on a random dataset $X \in \mathbb{R}^{d_0 \times n}$ whose entries are sampled iid Gaussian with mean 0 and variance 1. This gives us an activation region corresponding to the network evaluated at $X$, and we record the rank of the Jacobian of that matrix. For each choice of $d_0, d_1$, and $n$, we run 100 trials and record the fraction of them which resulted in a Jacobian of full rank. The results are shown in Figures 2 and 3.

For different scalings of $n$ and $d_0$, we observe different minimal widths $d_1$ needed for the Jacobian to achieve full rank with high probability. Figure 2 suggests that the minimum value of $d_1$ needed to achieve full rank increases linearly in the dataset size $n$, and that the slope of this linear dependence decreases as the input dimension $d_0$ increases. This is exactly the behavior predicted by Theorem 2, which finds full rank regions for $d_1 \gtrsim \frac{n}{d_0}$. Figure 3 operates in the regime $d_0 \sim n$, and shows that the necessary hidden dimension $d_1$ remains constant in the dataset size $n$. This is again consistent with Theorem 5, whose bounds depend only on the ratio $\frac{n}{d_0}$. Further supporting experiments, including those involving real-world data, are provided in Appendix E.

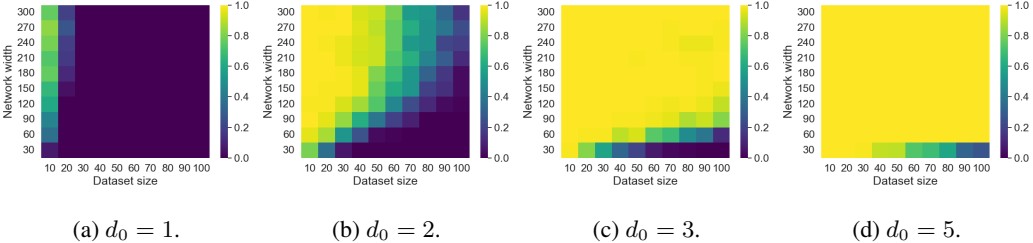

(a) $d_0 = 1$.      (b) $d_0 = 2$.      (c) $d_0 = 3$.      (d) $d_0 = 5$.

Figure 2: The probability of the Jacobian being of full rank from a random initialization for various values of $d_1$ and $n$, where the input dimension $d_0$ is left fixed.

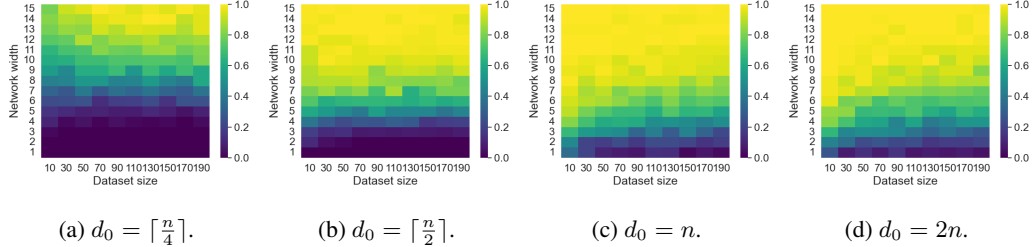

(a) $d_0 = \lceil \frac{n}{4} \rceil$.      (b) $d_0 = \lceil \frac{n}{2} \rceil$.      (c) $d_0 = n$.      (d) $d_0 = 2n$.

Figure 3: The probability of the Jacobian being of full rank for various values of $d_1$ and $n$, where the input dimension $d_0$ scales linearly in the number of samples $n$.

## 8 CONCLUSION

We studied the loss landscape of two-layer ReLU networks in the mildly overparameterized regime by identifying a large fraction of parameter space in which the loss function is well behaved. We showed that most activation regions have no bad differentiable local minima by using random matrix theory and combinatorial techniques to determine that these regions have a full rank Jacobian. For the one-dimensional input case, we further proved that most regions contain a high-dimensional set of global minimizers. Taking a qualitative perspective to count regions with beneficial properties allowed us to make rigorous the idea that most of the loss landscape is favorable to optimization. This combinatorial approach allowed us to prove results independent of the specific choice of initialization of parameters, or on the distribution of the dataset.

There are a few directions our theoretical results do not directly address. The fact that most regions of the optimization landscape are favorable suggests that randomly initialized gradient descent will land in a favorable region and avoid bad local minima on the interior of regions, but this needs to be made precise. We obtain our strongest results for the one-dimensional input case, where we have a concrete grasp of the possible activation patterns; the case $1 < d_1 < n$ remains open. Our choice of architecture for most of our results is also limited to two-layer networks, though we describe an extension to deep networks in Appendix H. We leave these questions to future work, and hope that our contributions inspire further exploration to address them.

**Reproducibility statement** The computer implementation of the scripts needed to reproduce our experiments can be found at `https://anonymous.4open.science/r/loss-landscape-4271`.

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

# A  DETAILS ON COUNTING ACTIVATION REGIONS WITH NO BAD LOCAL MINIMA

We provide the proofs of the results presented in Section 3.

*Proof of Lemma 4.* Suppose that $\lambda^{(1)}, \ldots, \lambda^{(n)} \in \mathbb{R}$. Since the entries of $v$ are nonzero, the following are equivalent:

$$\sum_{i=1}^{n} \lambda^{(i)} (v \odot a^{(i)}) \otimes x^{(i)} = 0$$

$$\sum_{i=1}^{n} \lambda^{(i)} (v \odot a^{(i)})_k x_j^{(i)} = 0 \quad \forall j \in [d_0], k \in [d_1]$$

$$\sum_{i=1}^{n} \lambda^{(i)} v_k a_k^{(i)} x_j^{(i)} = 0 \quad \forall j \in [d_0], k \in [d_1]$$

$$\sum_{i=1}^{n} \lambda^{(i)} a_k^{(i)} x_j^{(i)} = 0 \quad \forall j \in [d_0], k \in [d_1]$$

$$\sum_{i=1}^{n} \lambda^{(i)} a^{(i)} \otimes x^{(i)} = 0.$$

So the kernel of the matrix whose $i$-th row is $(v \odot a^{(i)}) \otimes x^{(i)}$ is equal to the kernel of the matrix whose $i$-th row is $a^{(i)} \otimes x^{(i)}$. It follows that

$$\mathrm{rank}(\{(v \odot a^{(i)}) \otimes x^{(i)} : i \in n\}) = \mathrm{rank}(\{a^{(i)} \otimes x^{(i)} : i \in [n]\}).$$

$\square$

*Proof of Theorem 5.* Suppose that $d_0 d_1 \geq n$. Consider a matrix $A \in \{0,1\}^{d_1 \times n}$, which corresponds to one of the theoretically possible activation pattern of all $d_1$ units across all $n$ input examples, and denote its columns by $a^{(j)} \in \{0,1\}^{d_1}, j = 1, \ldots, n$. For any given input dataset $X$, the function $F$ is piecewise linear in $W$. More specifically, on each activation region $\mathcal{S}_X^A$, $F(\cdot, v, X)$ is a linear map in $W$ of the form

$$F(W, v, x^{(j)}) = \sum_{i=1}^{d_1} \sum_{k=1}^{d_0} v_i A_{ij} W_{ik} x_k^{(j)} \qquad (j \in [n]).$$

So on $\mathcal{S}_X^A$ the Jacobian of $F(\cdot, X)$ is given by the Khatri-Rao (columnwise Kronecker) product

$$\nabla_W F(W, v, X) = ((v \odot a^{(1)}) \otimes x^{(1)}, \ldots, (v \odot a^{(n)}) \otimes x^{(n)}). \qquad (3)$$

In particular, $\nabla_W F$ is of full rank on $\mathcal{S}_X^A$ exactly when the set

$$\{(v \odot a^{(i)}) \otimes x^{(i)} : i \in [n]\}$$

consists of linearly independent elements of $\mathbb{R}^{d_1 \times d_0}$, since $d_0 d_1 \geq n$. By Lemma 4, this is equivalent to the set

$$\{a^{(i)} \otimes x^{(i)} : i \in [n]\}$$

consisting of linearly independent vectors, or in another words, the Khatri-Rao product $A * X$ having full rank.

For a given $A \in \{0,1\}^{d_1 \times n}$ consider the set

$$\mathcal{J}^A := \{X \in \mathbb{R}^{d_0 \times n} : A * X \text{ is of full rank}\}.$$

The expression $A * X$ corresponds to $\nabla_W F(W, v, X)$ in the case that $W \in \mathcal{S}_X^A$. Suppose that $d_1$ is large enough that $d_1 \geq \frac{n}{d_0}$. We will show that for most $A \in \{0,1\}^{d_1 \times n}$, $\mathcal{J}^A$ is non-empty and in fact contains almost every $X$. We may partition $[n]$ into $r := \lceil \frac{n}{d_1} \rceil$ subsets $S_1, S_2, \ldots, S_r$ such that

$|S_k| \le d_1$ for all $k \in [r]$ and partition the set of columns of $A$ accordingly into blocks $(a^{(s)})_{s \in S_k}$ for all $k \in [r]$. Let us form a $d_1 \times |S_k|$ matrix $M$ whose columns are the $a^{(s)}, s \in S_k$. We will use a probabilistic argument. For this, consider the entries of $M$ as being iid Bernoulli random variables with parameter $1/2$. We may extend $M$ to a $d_1 \times d_1$ random matrix $\tilde{M}$ whose entries are iid Bernoulli random variables with parameter $1/2$. By Theorem 3, $\tilde{M}$ will be singular with probability at most

$$\left( \frac{1}{\sqrt{2}} + o(1) \right)^{d_1} \le C_1 \cdot 0.72^{d_1},$$

where $C_1$ is a universal constant. Whenever $\tilde{M}$ is nonsingular, the vectors $a^{(s)}, s \in S_k$ are linearly independent. Using a simple union bound, we have

$$\mathbb{P}\left( (a^{(s)})_{s \in S_k} \text{ are linearly independent for all } k \in [r] \right) \ge 1 - rC_1(0.72)^{d_1}.$$

Now suppose that $(a^{(s)})_{s \in S_k}$ are linearly independent for all $k \in [r]$. Let $e_1, \ldots, e_{d_0}$ be the standard basis of $\mathbb{R}^{d_0}$. Since $r \le d_0$, there exists $X \in \mathbb{R}^{d_0 \times n}$ such that $x^{(i)} = e_k$ whenever $i \in S_k$. For such an $X$, we claim that the set

$$\{a^{(i)} \otimes x^{(i)} : i \in [n]\}$$

consists of linearly independent elements of $\mathbb{R}^{d_1 \times d_0}$. To see this, suppose that there exist $\alpha_1, \ldots, \alpha_n$ such that

$$\sum_{i=1}^{n} \alpha_i(a^{(i)} \otimes x^{(i)}) = 0.$$

Then

$$\begin{aligned}
0 &= \sum_{k=1}^{r} \sum_{i \in S_k} \alpha_i(a^{(i)} \otimes x^{(i)}) \\
&= \sum_{k=1}^{r} \sum_{i \in S_k} \alpha_i(a^{(i)} \otimes e_k) \\
&= \sum_{k=1}^{r} \left( \sum_{i \in S_k} \alpha_i a^{(i)} \right) \otimes e_k.
\end{aligned}$$

The above equation can only hold if

$$\sum_{i \in S_k} \alpha_i a^{(i)} = 0, \quad \text{for all } k \in [r].$$

By the linear independence of $(a^{(i)})_{i \in S_k}$, this implies that $\alpha_i = 0$ for all $i \in [n]$. This shows that the elements $a^{(i)} \otimes x^{(i)}$ are linearly independent for a particular $X$ whenever the $(a^{(i)})_{i \in S_k}$ are all linearly independent. In other words, $\mathcal{J}^A$ is non-empty with probability at least $1 - C_1 r(0.72^{d_1})$ when the activation region is chosen uniformly at random.

Let us define

$$\mathcal{A} := \{A \in \{0,1\}^{d_1 \times n} : \mathcal{J}^A \ne \emptyset\}.$$

We have shown that if $A$ is chosen uniformly at random from $\{0,1\}^{d_1 \times n}$, then $A \in \mathcal{A}$ with high probability. Note that $\mathcal{J}^A$ is defined in terms of polynomials of $X$ not vanishing, so $\mathcal{J}^A$ is the complement of a Zariski-closed subset of $\mathbb{R}^{d_0 \times n}$. Let

$$\mathcal{J} := \cap_{A \in \mathcal{A}} \mathcal{J}_A.$$

Then $\mathcal{J}$ is a Zariski-open set of full measure, since it is a finite intersection of non-empty Zariski-open sets (which are themselves of full measure by Lemma 1). If $X \in \mathcal{J}$ and $A \in \mathcal{A}$, then $\nabla_W F(W, X)$ is of full rank for all $W \in \mathcal{S}_X^A$, and therefore all local minima in $\mathcal{S}_X^A$ will be global minima by Lemma 2. So if we take $X \in \mathcal{J}$ and $d_1$ such that

$$d_1 \ge \frac{\log \left( \frac{C_1(n+1)}{d_0 \epsilon} \right)}{\log \left( \frac{1}{0.72} \right)}$$

and choose $A \in \{0,1\}^{d_1 \times n}$ uniformly at random, then with probability at least

$$1 - C_1 r(0.72^{d_1}) \geq 1 - C_1 r \left( \frac{d_0 \epsilon}{C_1(n+1)} \right)$$
$$\geq 1 - \epsilon,$$

$\mathcal{S}_X^A$ will have no bad local minima. We rephrase this as a combinatorial result: if $d_1$ satisfies the same bounds above, then for generic datasets $X$ we have the following: for all but at most $\lceil \epsilon 2^{d_0 d_1} \rceil$ activation regions $\mathcal{S}_X^A$, $\mathcal{S}_X^A$ has no bad local minima. The theorem follows. $\qquad \square$

The argument that we used to prove that the Jacobian is full rank for generic data can be applied to arbitrary binary matrices. The proof is exactly the same as the sub-argument in the proof of Theorem 5.

**Lemma 14.** *For generic datasets $X \in \mathbb{R}^{d_0 \times n}$, the following holds. Let $a^{(i)} \in \{0,1\}^{d_1}$ for $i \in [n]$. Suppose that there exists a partition of $[n]$ into $r \leq d_0$ subsets $S_1, \cdots, S_r$ such that for all $k \in [r]$,*

$$\mathrm{rank}(\{a^{(i)} : i \in S_k\}) = |S_k|.$$

*Then*

$$\mathrm{rank}(\{a^{(i)} \otimes x^{(i)} : i \in [n]\}) = n.$$

## B    DETAILS ON COUNTING NON-EMPTY ACTIVATION REGIONS

We provide the proofs of the results presented in Section 4. The subdivisions of parameter space by activation properties of neurons is classically studied in VC-dimension computation (Cover, 1965; Sakurai, 1998; Anthony & Bartlett, 1999). This has similarities to the analysis of linear regions in input space for neural networks with piecewise linear activation functions (Montúfar et al., 2014).

*Proof of Proposition 6.* Consider the map $\mathbb{R}^{d_0} \to \mathbb{R}^n; w \mapsto [w^T X]_+$ that takes the input weights $w$ of a single ReLU to its activation values over $n$ input data points given by the columns of $X$. This can equivalently be interpreted as a map taking an input vector $w$ to the activation values of $n$ ReLUs with input weights given by the columns of $X$. The linear regions of the latter correspond to the (full-dimensional) regions of a central arrangement of $n$ hyperplanes in $\mathbb{R}^{d_0}$ with normals $x^{(1)}, \ldots, x^{(n)}$. Denote the number of such regions by $N_X$. If the columns of $X$ are in general position in a $d$-dimensional linear space, meaning that they are contained in a $d$-dimensional linear space and any $d$ columns are linearly independent, then

$$N_X = N_{d,n} := 2 \sum_{k=0}^{d-1} \binom{n-1}{k}. \tag{4}$$

This is a classic result that can be traced back to the work of Schläfli (1950), and which also appeared in the discussion of linear threshold devices by Cover (1964).

Now for a layer of $d_1$ ReLUs, each unit has its parameter space $\mathbb{R}^{d_0}$ subdivided by an equivalent hyperplane arrangement that is determined solely by $X$. Since all units have individual parameters, the arrangement of each unit essentially lives in a different set of coordinates. In turn, the overall arrangement in the parameter space $\mathbb{R}^{d_0 \times d_1}$ of all units is a so-called product arrangement, and the number of regions is $(N_X)^{d_1}$. This conclusion is sufficiently intuitive, but it can also be derived from Zaslavsky (1975, Lemma 4A3). If the input data $X$ is in general position in a $d$-dimensional linear subspace of the input space, then we can substitute equation 4 into $(N_X)^{d_1}$ and obtain the number of regions stated in the proposition. $\qquad \square$

We are also interested in the specific identity of the non-empty regions; that is, the sign patterns that are associated with them. As we have seen above, the set of sign patterns of a layer of units is the $d_1$-Cartesian power of the set of sign patterns of an individual unit. Therefore, it is sufficient to understand the set of sign patterns of an individual unit; that is, the set of dichotomies that can be computed over the columns of $X$ by a bias-free simple perceptron $x \mapsto \mathrm{sgn}(w^T x)$. Note that this subsumes networks with biases as the special case where the first row of $X$ consists of ones, in which

case the first component of the weight vector can be regarded as the bias. Let $L_X$ be the $N_X \times n$ matrix whose $N_X$ rows are the different possible dichotomies $(\operatorname{sgn} w^T x^{(1)}, \ldots, \operatorname{sgn} w^T x^{(n)}) \in \{-1, +1\}^n$. If we extend the definition of dichotomies to allow not only $+1$ and $-1$ but also zeros for the case that data points fall on the decision boundary, then we obtain a matrix $L_X$ that is referred to as the *oriented matroid* of the vector configuration $X$, and whose rows are referred to as the *covectors* of $X$ (Björner et al., 1999). This is also known as the list of sign sequences of the faces of the hyperplane arrangement.

To provide more intuitions for Proposition 7, we give a self-contained proof below.

*Proof of Proposition 7.* For each unit, the parameter space $\mathbb{R}^{d_0}$ is subdivided by an arrangement of $n$ hyperplanes with normals given by $x^{(j)}$, $j \in [n]$. A weight vector $w$ is in the interior of the activation region with pattern $a \in \{0, 1\}^n$ if and only if $(2a_j - 1)w^T x^{(j)} > 0$ for all $j \in [n]$. Equivalently,

$$w^T x^{(j)} > w^T 0 \quad \text{for all } j \text{ with } a_j = +1$$
$$w^T 0 > w^T x^{(j')} \quad \text{for all } j' \text{ with } a'_j = 0.$$

This means that $w$ is a point where the function $w \mapsto w^T \sum_{j \colon a_j = 1} x^{(j)}$ attains the maximum value among of all linear functions with gradients given by sums of $x^{(j)}$s, meaning that at $w$ this function attains the same value as

$$\psi \colon w \mapsto \sum_{j \in [n]} \max\{0, w^T x^{(j)}\} = \max_{S \subseteq [n]} w^T \sum_{j \in S} x^{(j)}. \tag{5}$$

Dually, the linear function $x \mapsto w^T x$ attains its maximum over the polytope

$$P = \operatorname{conv}\{\sum_{j \in S} x^{(j)} \colon S \subseteq [n]\} \tag{6}$$

precisely at $\sum_{j \colon a_j = 1} x^{(j)}$. In other words, $\sum_{j \colon a_j = 1} x^{(j)}$ is an extreme point or vertex of $P$ with a supporting hyperplane with normal $w$. For a polytope $P \subseteq \mathbb{R}^{d_0}$, the normal cone of $P$ at a point $x \in P$ is defined as the set of $w \in \mathbb{R}^{d_0}$ such that $w^T x \geq w^T x'$ for all $x' \in P$. For any $S \subseteq [n]$ let us denote by $\mathbf{1}_S \in \{0, 1\}^{[n]}$ the vector with ones at components in $S$ and zeros otherwise. Then the above discussion shows that the activation region $\mathcal{S}_X^a$ with $a = \mathbf{1}_S$ is the interior of the normal cone of $P$ at $\sum_{j \in S} x^{(j)}$. In particular, the activation region is non empty if and only if $\sum_{j \in S} x^{(j)}$ is a vertex of $P$.

To conclude, we show that $P$ is a Minkowski sum of line segments,

$$P = \sum_{j \in [n]} P_j, \quad P_i = \operatorname{conv}\{0, x^{(j)}\}.$$

To see this note that $x \in P$ if and only if there exist $\alpha_S \geq 0$, $\sum_S \alpha_S = 1$ with

$$\begin{aligned}
x &= \sum_{S \subseteq [n]} \alpha_S \sum_{j \in S} x^{(j)} \\
&= \sum_{j \in [n]} \sum_{S \colon j \in S} \alpha_S x^{(j)} \\
&= \sum_{j \in [n]} [(\sum_{S \colon j \notin S} \alpha_S) 0 + (\sum_{S \colon j \in S} \alpha_S) x^{(j)}] \\
&= \sum_{j \in [n]} [(1 - \beta_j) 0 + \beta_j x^{(j)}] = \sum_{j \in [n]} z^{(j)},
\end{aligned}$$

where $\beta_j = \sum_{S \colon j \in S} \alpha_S$ and $z^{(j)} = [(1 - \beta_j) 0 + \beta_j x^{(j)}]$. Thus, $x \in P$ if and only if $x$ is a sum of points $z^{(j)} \in P_j = \operatorname{conv}\{0, x^{(j)}\}$, meaning that $P = \sum_j P_j$ as was claimed. $\square$

The polytope equation 6 may be regarded as the Newton polytope of the piecewise linear convex function equation 5 in the context of tropical geometry (Joswig, 2021). This perspective has been

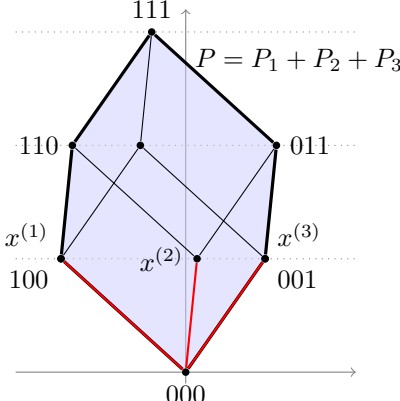

Figure 4: Illustration of Proposition 7. The polytope $P$ for a ReLU on three data points $x^{(1)}, x^{(2)}, x^{(3)}$ is the Minkowski sum of the line segments $P_i = \text{conv}\{0, x^{(i)}\}$ highlighted in red. The activation regions in parameter space are the normal cones of $P$ at its different vertices. Hence the vertices correspond to the non-empty activation regions. These are naturally labeled by vectors $\mathbf{1}_S$ that indicate which $x^{(i)}$ are added to produce the vertex and record the activation patterns.

used to study the linear regions in input space for ReLU (Zhang et al., 2018) and maxout networks (Montúfar et al., 2022).

We note that each vertex of a Minkowski sum of polytopes $P = P_1 + \cdots + P_n$ is a sum of vertices of the summands $P_j$, but not every sum of vertices of the $P_j$ results in a vertex of $P$. Our polytope $P$ is a sum of line segments, which is a type of polytope known as a zonotope. Each vertex of $P$ takes the form $v = \sum_{j \in [n]} v_j$, where each $v_j$ is a vertex of $P_j$, either the zero vertex 0 or the nonzero vertex $x^{(j)}$, and is naturally labeled by a vector $\mathbf{1}_S \in \{0, 1\}^n \cong 2^{[n]}$ that indicates the $j$s for which $v_j = x^{(j)}$. A zonotope can be interpreted as the image of a cube by a linear map; in our case it is the projection of the $n$-cube $\text{conv}\{a \in \{0, 1\}^n\} \subseteq \mathbb{R}^n$ into $\mathbb{R}^{d_0}$ by the matrix $X \in \mathbb{R}^{d_0 \times n}$. The situation is illustrated in Figure 4.

**Example 15** (Non-empty activation regions for 1-dimensional inputs). In the case of one-dimensional inputs and units with biases, the parameter space of each unit is $\mathbb{R}^2$. We treat the data as points $x^{(1)}, \ldots, x^{(n)} \in 1 \times \mathbb{R}$, where the first coordinate is to accommodate the bias. For generic data (i.e., no data points on top of each other), the polytope $P = \sum_{j \in [d_1]} \text{conv}\{0, x^{(j)}\}$ is a polygon with $2n$ vertices. The vertices have labels $\mathbf{1}_S$ indicating, for each $k = 0, \ldots, n$, the subsets $S \subseteq [n]$ containing the largest respectively smallest $k$ elements in the dataset with respect to the non-bias coordinate.

## C DETAILS ON COUNTING NON-EMPTY ACTIVATION REGIONS WITH NO BAD MINIMA

We provide the proofs of the statements in Section 5.

*Proof of Lemma 9.* First we establish that the rows of the matrices $A$ corresponding to non-empty activation regions must be step vectors. To this end, assume $A \in \mathcal{S}_X^A$ and for an arbitrary row $i \in [d_1]$ let $A_{i1} = \alpha \in \{0, 1\}$. If $A_{ij} = \alpha$ for all $j \in [n]$, then the $i$-th row of $A$ is a step vector equal to either $\xi^{(n+1,0)}$ or $\xi^{(n+1,1)}$. Otherwise, there exists a minimal $k \in [n+1]$ such that $A_{ik} = 1 - \alpha$. We proceed by contradiction to prove in this setting that $A_{ij} = 1 - \alpha$ for all $j \geq k$. Suppose there exists a $j > k$ such that $A_{ij} = \alpha$, then as $A \in \mathcal{S}_X^A$ and $x^{(k)} < x^{(j)}$ this implies that the following two inequalities are simultaneously satisfied,

$$0 < (2A_{ik} - 1)(w^{(i)}x^{(k)} + b^{(i)}) = (1 - 2\alpha)(w^{(i)}x^{(k)} + b^{(i)}) < (1 - 2\alpha)(w^{(i)}x^{(j)} + b^{(i)}),$$

$$0 < (2A_{ij} - 1)(w^{(i)}x^{(j)} + b^{(i)}) = (2\alpha - 1)(w^{(i)}x^{(j)} + b^{(i)}) = -(1 - 2\alpha)(w^{(i)}x^{(j)} + b^{(i)}).$$

However, these inequalities clearly contradict one another and therefore we conclude $A_{i,\cdot}$ is a step vector equal to either $\xi^{(k,0)}$ or $\xi^{(k,1)}$

Conversely, assume the rows of $A$ are all step vectors. We proceed to prove $\mathcal{S}_X^A$ is non-empty under this assumption by constructing $(w, b)$ such that $\text{sign}(w^{(i)}x^{(j)} + b^{(i)}) = A_{ij}$ for all $j \in [n]$ and any row $i \in [d_1]$. To this end, let $A_{i1} = \alpha \in \{0, 1\}$. First, consider the case where $A_{ij} = \alpha$ for all $j \in [n]$. If $\alpha = 0$ then with $w^{(i)} = 1$ and $b^{(i)} = -2|x^{(n)}| - 1$ it follows that

$$w^{(i)}x^{(j)} + b^{(i)} < x^{(n)} + b^{(i)} < -|x^{(n)}| - 1$$

for all $j \in [n]$. If $\alpha = 1$, then with $w^{(i)} = 1$ and $b^{(i)} = 2|x^{(i)}| + 1$ we have

$$w^{(i)}x^{(j)} + b^{(i)} > w^{(i)}x^{(1)} + b^{(i)} > |x^{(1)}| + 1$$

for all $j \in [n]$. Otherwise, suppose $A_{i,\cdot}$ is not constant: then by construction there exists a $k \in \{2, \cdots, n\}$ such that $A_{ij} = \alpha$ for all $j \in [1, k-1]$ and $A_{ij} = 1 - \alpha$ for all $j \in [k, n]$. Letting

$$w^{(i)} = (2\alpha - 1),$$
$$b^{(i)} = -(2\alpha - 1)\left(\frac{x^{(k-1)} + x^{(k)}}{2}\right),$$

then for any $j \in [n]$

$$\text{sgn}(w^{(i)}x^{(j)} + b^{(i)}) = \text{sgn}\left((2\alpha - 1)\left(x^{(j)} - \frac{x^{(k-1)} + x^{(k)}}{2}\right)\right) = A_{ij}.$$

Thus, for any $A$ with step vector rows, given a dataset consisting distinct one dimensional points we can construct network whose preactivations correspond to the activation pattern encoded by $A$.

In summary, given a fixed, distinct one dimensional dataset we have established a one-to-one correspondence between the non-empty activation regions and the set of binary matrices whose rows are step vectors. For convenience we refer to these as row-step matrices. As there are $2n$ step vector rows of dimension $n$ and $d_1$ rows in $A$, then there are $(2n)^{d_1}$ binary row-step matrices and hence $(2n)^{d_1}$ non-empty activation regions. $\qquad\square$

*Proof of Lemma 11.* By the union bound,

$$\mathbb{P}([n] \not\subseteq \{C_1, \ldots, C_d\}) \leq \sum_{j=1}^{n} \mathbb{P}(j \notin \{C_1, \cdots, C_d\})$$
$$= \sum_{j=1}^{n} (1 - \delta)^d$$
$$\leq \sum_{j=1}^{n} e^{-\delta d}$$
$$= ne^{-\delta d}.$$

So if $d > \frac{1}{\delta}\log(\frac{n}{\epsilon})$,

$$\mathbb{P}([n] \subseteq \{C_1, \ldots, C_d\}) \geq 1 - ne^{-\delta d}$$
$$\geq 1 - \epsilon.$$

This concludes the proof. $\qquad\square$

*Proof of Theorem 10.* Consider a dataset $(X, y)$ consisting of distinct data points. Without loss of generality we may index these points such that

$$x^{(1)} < x^{(2)} < \cdots < x^{(n)}.$$

Now consider a matrix $A \in \{0,1\}^{d_1 \times n}$ whose rows are step vectors and which therefore corresponds to a non-empty activation region by Lemma 9. As in the proof of Theorem 5, denote its columns by $a^{(j)} \in \{0,1\}^{d_1}$ for $j \in [n]$. On $\mathcal{S}_X^A$, the Jacobian of $F$ with respect to $b$ is given by

$$\nabla_b F(w, b, v, X) = (v \odot a^{(1)}, \ldots, v \odot a^{(n)}).$$

We claim that if $A$ has full rank, then $\nabla_b F(w, b, v, X)$ has full rank for all $w, b, v, X$ with the entries of $v$ nonzero. To see this, suppose that $A$ is of full rank, and that

$$\sum_{j=1}^n \alpha_j (v \odot a^{(j)}) = 0$$

for some $\alpha_1, \ldots, \alpha_n \in \mathbb{R}$. Then for all $i \in [d_1]$,

$$\sum_{j=1}^n \alpha_j v_i A_{ij} = 0,$$

$$\sum_{j=1}^n \alpha_i A_{ij} = 0.$$

But $A$ is of full rank, so this implies that $\alpha_i = 0$ for all $i \in [n]$. As a result, if $A$ is full rank then $\nabla_b F(w, b, v, X)$ is full rank, and in particular $\nabla_{(w,b,v)} F(w, b, v, X)$ is rank. Therefore, to show most non-empty activation regions have no bad local minima, it suffices to show most non-empty activation regions are defined by a full rank binary matrix $A$.

To this end, if $A$ is a binary matrix with step vector rows, we say that $A$ is *diverse* if it satisfies the following properties:

1. For all $k \in [n]$, there exists $i \in [d_1]$ such that $A_{i,\cdot} \in \{\xi^{(k,0)}, \xi^{(k,1)}\}$.

2. There exists $i \in [d_1]$ such that $A_{i,\cdot} = \xi^{(1,1)}$.

We proceed by i) showing all diverse matrices are of full rank and then ii) non-empty activation regions are defined by a diverse matrix. Suppose $A$ is diverse and denote the span of the rows of $A$ by row$(A)$. Then $\xi^{(1,1)} = (1, \ldots, 1) \in$ row$(A)$ and for each $k \in [n]$ either $\xi^{(k,0)}$ or $\xi^{(k,1)}$ is in row$(A)$. Observe if $\xi^{(k,0)} \in$ row$(A)$ then $1 - \xi^{(k,0)} = \xi^{(k,1)} \in$ row$(A)$, therefore for all $k \in [n]$ $\xi^{(k,1)} \in$ row$(A)$. As the set of vectors

$$\{\xi^{(k,1)} : k \in [n]\}$$

forms a basis of $\mathbb{R}^n$ we conclude that all diverse matrices are full rank.

Now we show most binary matrices with step vector rows are diverse. Let $A$ be a random binary matrix whose rows are selected mutually iid from the set of all step vectors. For $i \in [d_1]$, let $C_i \in [n+1]$ be defined as follows: if $A_{i,\cdot} \in \{\xi^{(k,0)}, \xi^{(k,1)}\}$ for some $k \in \{2, 3, \ldots, n\}$, we define $C_i = k$; if $A_{i,\cdot} = \xi^{(1,1)}$, then we define $C_i = 1$; otherwise, we define $C_i = n + 1$. By definition $A$ is diverse if and only if

$$[n] \subseteq \{C_1, \ldots, C_{d_1}\}.$$

As the rows of $A$ are chosen uniformly at random from the set of all step vectors, the $C_i$ are iid and

$$\mathbb{P}(C_1 = k) \geq \frac{1}{2n}$$

for all $k \in [n]$, then by Lemma 11, if $d \geq 2n \log(\frac{n}{\epsilon})$,

$$\mathbb{P}(A \text{ is diverse}) = \mathbb{P}([n] \subseteq \{C_1, \ldots, C_{d_1}\})$$
$$\geq 1 - \epsilon.$$

This holds for a randomly selected matrix with step vector rows. Translating this into a combinatorial statement, we see for all but at most a fraction $\epsilon$ of matrices with step vector rows are diverse. Furthermore, in each activation region $\mathcal{S}_X^A$ corresponding to a diverse $A$ the Jacobian is full rank and every differentiable critical point of $L$ is a global minimum. $\square$

*Proof of Theorem 12.* Let $\epsilon > 0$. We define the following two sets of neurons based on the sign of their output weight,

$$\mathcal{D}_1 = \{i \in [d_1] : v^{(i)} > 0\},$$
$$\mathcal{D}_0 = \{i \in [d_1] : v^{(i)} < 0\}.$$

Suppose that $|\mathcal{D}_1|, |\mathcal{D}_0| \geq 2n \log(\frac{2n}{\epsilon})$. Furthermore, without loss of generality, we index the points in the dataset such that

$$x^{(1)} < x^{(2)} < \cdots < x^{(n)}.$$

Now consider a matrix $A \in \{0,1\}^{d_1 \times n}$ whose rows are step vectors, then by Lemma 9 $A$ corresponds to a non-empty activation region. We say that $A$ is *complete* if for all $k \in \{0, \ldots, n-1\}$ and $\beta \in \{0, 1\}$ there exists $i \in [d_1]$ such that $A_{i,\cdot} = \xi^{(k,1)}$ and $\mathrm{sgn}(v^{(i)}) = \beta$. We first show that if $A$ is complete, then there exists a global minimizer in $\mathcal{S}_X^A$. Consider the linear map $\varphi : \mathbb{R}^{d_1} \times \mathbb{R}^{d_1} \to \mathbb{R}^{1 \times n}$ defined by

$$\varphi(w,b) = \left[ \sum_{i=1}^{d_1} v^{(i)}(2A_{ij} - 1)(w^{(i)} x^{(j)} + b^{(i)}) \right]_{j=1,\ldots,n}.$$

Note for $(w, b) \in \mathcal{S}_X^A$ then $\varphi(w, b) = F(w, b, v, X)$. As proved in Lemma 16, for every $z \in \mathbb{R}^{1 \times n}$ there exists $(w, b) \in \mathcal{S}_X^A$ such that $F(w, b, v, X) = z$. In particular, this means that $\varphi$ is surjective and therefore $\mathcal{S}_X^A$ contains zero-loss global minimizers. Define

$$\mathcal{V}_y = \{(w,b) \in \mathbb{R}^{d_1} \times \mathbb{R}^{d_1} : \varphi(w,b) = y\},$$

then $\mathcal{G}_{X,y} \cap \mathcal{S}_X^A = \mathcal{V}_y \cap \mathcal{S}_X^A$ and by the rank-nullity theorem

$$\dim(\mathcal{V}_y) = \dim(\varphi^{-1}(\{y\}))$$
$$= 2d_1 - n.$$

We therefore conclude that if $A$ is complete, then $\mathcal{G}_{X,y} \cap \mathcal{S}_X^A$ is the restriction of a $(2d_1 - n)$-dimensional linear subspace to the open set $\mathcal{S}_X^A$. This is equivalent to $\mathcal{G}_{X,y} \cap \mathcal{S}_X^A$ being an affine set of codimension $n$ as claimed.

To prove the desired result it therefore suffices to show that most binary matrices with step vector rows are complete. Let $A$ be a random binary matrix whose rows are selected mutually iid uniformly at random from the set of all step vectors. For $\beta \in \{0, 1\}$ and $i \in \mathcal{D}_\beta$ let $C_{\beta,i} \in [n]$ be defined as follows: if $A_{i,\cdot} = \xi^{(k-1,1)}$ for some $k \in [n]$ then $C_{\beta,i} = k$, otherwise $C_{\beta,i} = n+1$. Observe that $A$ is complete if and only if

$$[n] \subseteq \{C_{0,i} : i \in \mathcal{D}_0\} \text{ and } [n] \subseteq \{C_{1,i} : i \in \mathcal{D}_1\}.$$

Since there are $2n$ step vectors,

$$\mathbb{P}(C_{\beta,i} = k) = \frac{1}{2n}$$

for all $k \in [n]$. So by the union bound and Lemma 11,

$$\mathbb{P}(A \text{ is complete}) = \mathbb{P}([n] \subseteq \{C_{0,i} : i \in \mathcal{D}_0\}) \text{ and } [n] \subseteq \{C_{1,i} : i \in \mathcal{D}_1\})$$
$$\geq 1 - \mathbb{P}([n] \not\subseteq \{C_{0,i} : i \in \mathcal{D}_0\}) - \mathbb{P}([n] \not\subseteq \{C_{1,i} : i \in \mathcal{D}_1\})$$
$$\geq 1 - \frac{\epsilon}{2} - \frac{\epsilon}{2}$$
$$= 1 - \epsilon.$$

As this holds for a matrix with step vectors chosen uniformly at random, it therefore follows that all but at most a fraction $\epsilon$ of such matrices are complete. This concludes the proof. $\square$

**Lemma 16.** *Let $A \in \{0,1\}^{d_1 \times n}$ be complete, and let $X \in \mathbb{R}^{d_0 \times n}$ be an input dataset with distinct points. Then for any $y \in \mathbb{R}^{1 \times n}$, there exists $(w, b) \in \mathcal{S}_X^A$ such that $F(w, b, v, X) = y$.*

*Proof.* If $A$ is complete then there exist row indices $i_{0,0}, i_{1,0}, \ldots, i_{n-1,0} \in [d_1]$ and $i_{0,1}, i_{1,1}, \ldots, i_{n-1,1} \in [d_1]$ such that for all $k \in \{0, \ldots, n-1\}$ and $\beta \in \{0, 1\}$ the $i_{k,\beta}$-th row of $A$ is equal to $\xi^{(k,1)}$ with $v^{(i_{k,\beta})} = \beta$. For $\beta \in \{0, 1\}$, we define

$$\mathcal{I}_\beta = \{i_{0,\beta}, \ldots, i_{n-1,\beta}\}.$$

We will construct weights $w^{(1)}, \ldots, w^{(d_1)}$ and biases $b^{(1)}, \ldots, b^{(d_1)}$ such that $(w, b) \in \mathcal{S}_X^A$ and $F(w, b, v, X) = y$. For $i \notin \mathcal{I}_0 \cup \mathcal{I}_1$, we define $w^{(i)}$ and $b^{(i)}$ arbitrarily such that

$$\operatorname{sgn}(w^{(i)} x^{(j)} + b^{(i)}) = A_{ij}$$

for all $j \in [n]$. Note by Lemma 9 that this is possible as each $A_{i,\cdot}$ is a step vector. Therefore, in order to show the desired result it suffices only to appropriately construct $w^{(i)}, b^{(i)}$ for $i \in \mathcal{I}_0 \cup \mathcal{I}_1$. To this end we proceed using the following sequence of steps.

1. First we separately consider the contributions to the output of the network coming from the *slack* neurons, those with index $i \notin \mathcal{I}_0 \cup \mathcal{I}_1$, and the *key* neurons, those with index $i \in \mathcal{I}_0 \cup \mathcal{I}_1$. For $j \in [n]$, we denote the residual of the target $y^{(j)}$ leftover after removing the corresponding output from the slack part of the network as $z^{(j)}$. We then recursively build a sequence of functions $(g^{(l)})_{l=0}^{n-1}$ in such a manner that $g^{(l)}(x^{(j)}) = z^{(j)}$ for all $j \leq n$.

2. Based on this construction, we select the parameters of the key neurons, i.e., $(w^{(i)}, b^{(i)})$ for $i \in \mathcal{I}_0 \cup \mathcal{I}_1$, and prove $(w, b) \in \mathcal{S}_X^A$.

3. Finally, using these parameters and the function $g^{(n-1)}$ we show $F(w, b, v, X) = y$.

**Step 1:** for $j \in [n]$ we define the residual

$$z^{(j)} = y^{(j)} - \sum_{i \notin \mathcal{I}_0 \cup \mathcal{I}_1} v^{(i)} \sigma(w^{(i)} x^{(j)} + b^{(i)}). \tag{7}$$

For $j \in \{0, \ldots, n-1\}$, consider the values $\beta^{(j)}, \tilde{b}^{(j)}, \tilde{w}^{(j)} \in \mathbb{R}$, and the functions $g^{(j)} : \mathbb{R} \to \mathbb{R}$, defined recursively across the dataset as

$$\beta^{(0)} = \operatorname{sgn}(z^{(1)})$$

$$\tilde{w}^{(0)} = 1$$

$$\tilde{b}^{(0)} = |z^{(1)}| - \tilde{x}^{(1)}$$

$$g^{(0)}(x) = (2\beta^{(0)} - 1)\sigma(\tilde{w}^{(0)} x + \tilde{b}^{(0)})$$

$$\beta^{(j)} = \operatorname{sgn}(z^{(j+1)} - g^{(j-1)}(x^{(j+1)})) \qquad (1 \leq j \leq n-1)$$

$$\tilde{w}^{(j)} = \frac{2|z^{(j+1)} - g^{(j-1)}(x^{(j+1)})|}{x^{(j+1)} - x^{(j)}} \qquad (1 \leq j \leq n-1)$$

$$\tilde{b}^{(j)} = -\frac{|z^{(j+1)} - g^{(j-1)}(x^{(j+1)})|(x^{(j+1)} + x^{(j)})}{x^{(j+1)} - x^{(j)}} \qquad (1 \leq j \leq n-1)$$

$$g^{(j)}(x) = \sum_{\ell=0}^{j} (2\beta^{(\ell)} - 1)\sigma(\tilde{w}^{(\ell)} x + \tilde{b}^{\ell}) \qquad (1 \leq j \leq n-1).$$

Observe for all $j \in \{1, \ldots, n-1\}$ that $\tilde{w}^{(j)} \geq 0$ and

$$\tilde{w}^{(j)} \left( \frac{x^{(j)} + x^{(j+1)}}{2} \right) + b^{(j)} = 0.$$

In particular, $\tilde{w}^{(j)} x^{(j')} + \tilde{b}^{(j)} < 0$ if $j' \leq j$ and $\tilde{w}^{(j)} x^{(j')} + b^{(j)} > 0$ otherwise. Moreover, for all $j' \in [n]$,

$$\tilde{w}^{(0)} x^{(j')} + b^{(0)} = x^{(j')} + |\tilde{y}^{(1)}| - x^{(1)}$$

$$> 0.$$

We claim that for all $j \in [n]$, $g^{(n-1)}(x^{(j)}) = z^{(j)}$. For the case $j = 1$, we compute

$$g^{(n-1)}(x^{(1)}) = \sum_{\ell=0}^{n-1} (2\beta^{(\ell)} - 1)\sigma(\tilde{w}^{(\ell)} x^{(1)} + \tilde{b}^{(\ell)})$$

$$= (2\beta^{(0)} - 1)\sigma(\tilde{w}^{(0)} x^{(1)} + \tilde{b}^{(0)})$$

$$= (2\operatorname{sgn}(\tilde{z}^{(1)}) - 1)\sigma(x^{(1)} + |\tilde{z}^{(1)}| - x^{(1)})$$

$$= \tilde{z}^{(1)}.$$

For $j \in \{2, \cdots, n-1\}$,

$$g^{(n-1)}(x^{(j)})$$

$$= \sum_{\ell=0}^{n-1} (2\beta^{(\ell)} - 1)\sigma(\tilde{w}^{(\ell)}x^{(j)} + \tilde{b}^{(\ell)})$$

$$= \sum_{\ell=0}^{j-1} (2\beta^{(\ell)} - 1)\sigma(\tilde{w}^{(\ell)}x^{(j)} + \tilde{b}^{(\ell)})$$

$$= (2\beta^{(j-1)} - 1)\sigma(\tilde{w}^{(j-1)}x^{(j)} + \tilde{b}^{(j-1)}) + g^{(j-2)}(x^{(j)})$$

$$= (2\beta^{(j-1)} - 1)(\tilde{w}^{(j-1)}x^{(j)} + \tilde{b}^{(j-1)}) + g^{(j-2)}(x^{(j)})$$

$$= (2\beta^{(j-1)} - 1)(\tilde{w}^{(j-1)}x^{(j)} + \tilde{b}^{(j-1)}) + g^{(j-2)}(x^{(j)})$$

$$= (2\beta^{(j-1)} - 1)\left( \frac{2|z^{(j)} - g^{(j-2)}(x^{(j)})|}{x^{(j)} - x^{(j-1)}}x^{(j)} - \frac{|z^{(j)} - g^{(j-2)}(x^{(j)})|(x^{(j)} + x^{(j-1)})}{x^{(j)} - x^{(j-1)}} \right)$$

$$\quad + g^{(j-2)}(x^{(j)})$$

$$= (2\beta^{(j-1)} - 1)|z^{(j)} - g^{(j-2)}(x^{(j)})| + g^{(j-2)}(x^{(j)})$$

$$= (z^{(j)} - g^{(j-2)}(x^{(j)})) + g^{(j-2)}(x^{(j)})$$

$$= z^{(j)}.$$

Hence, $g^{(n-1)}(x^{(j)}) = z^{(j)}$ for all $j \in [n]$.

**Step 2:** based on the construction above we define $(w^{(i)}, b^{(i)})$ for $i \in \mathcal{I}_0 \cup \mathcal{I}_1$. For $k \in \{0, \ldots, n-1\}$ and $\beta \in \{0, 1\}$, we define

$$w^{(i_{k,\beta})} = \frac{3 + (2\beta^{(k)} - 1)(2\beta - 1)}{2} \frac{\tilde{w}^{(k)}}{|v^{(i_{k,\beta})}|}$$

$$b^{(i_{k,\beta})} = \frac{3 + (2\beta^{(k)} - 1)(2\beta - 1)}{2} \frac{\tilde{b}^{(k)}}{|v^{(i_{k,\beta})}|}.$$

Now we show that this pair $(w, b)$ satisfies the desired properties. By construction, we have

$$\text{sgn}(w^{(i)}x^{(j)} + b^{(i)}) = A_{ij}$$

for $i \notin \mathcal{I}_0 \cup \mathcal{I}_1, j \in [n]$. If $i \in \mathcal{I}_0 \cup \mathcal{I}_1, j \in [n]$, then we can write $i = i_{k,\beta}$ for some $k \in \{0, \ldots, n-1\}$, $\beta \in \{0, 1\}$. Then

$$\text{sgn}(w^{(i)}x^{(j)} + b^{(i)}) = \text{sgn}\left( \frac{3 + (2\beta^{(k)} - 1)(2\beta - 1)}{2|v^{(i_{k,\beta})}|}(\tilde{w}^{(k)}x^{(j)} + \tilde{b}^{(k)}) \right)$$

$$= \text{sgn}(\tilde{w}^{(k)}x^{(j)} + \tilde{b}^{(k)})$$

$$= 1_{k<j}$$

$$= A_{ij},$$

where the second-to-last line follows from the construction of $\tilde{w}$ and $\tilde{b}$ and the last line follows from the fact that $A_{i,\cdot} = \xi^{(k,1)}$.

**Step 3:** finally, we show that $F(w, b, v, X) = y$. For $j \in [n]$,

$$F(w, b, x^{(j)}) = \sum_{i=1}^{d_1} v^{(i)}\sigma(w^{(i)}x^{(j)} + b^{(i)})$$

$$= \sum_{i \notin \mathcal{I}_0 \cup \mathcal{I}_1} v^{(i)}\sigma(w^{(i)}x^{(j)} + b^{(i)}) + \sum_{i \in \mathcal{I}_0 \cup \mathcal{I}_1} v^{(i)}\sigma(w^{(i)}x^{(j)} + b^{(i)})$$

$$= z^{(j)} - y^{(j)} + \sum_{k=0}^{n-1}\sum_{\beta=0}^{1} v^{(i_{k,\beta})}\sigma(w^{(i_{k,\beta})}x^{(j)} + b^{(i_{k,\beta})}),$$

where the last line follows from (7). By the definition of $w^{(i_k,\beta)}$ and $b^{(i_k,\beta)}$, this is equal to

$$z^{(j)} - y^{(j)} + \sum_{k=0}^{n-1} \sum_{\beta=0}^{1} \frac{v^{(i_k,\beta)}}{|v^{(i_k,\beta)}|} \frac{3 + (2\beta^{(k)} - 1)(2\beta - 1)}{2} \sigma(\tilde{w}^{(k)} x^{(j)} + \tilde{b}^{(k)}).$$

By construction $\operatorname{sgn}(v^{(i_k,\beta)}) = \beta$, so the above is equal to

$$y^{(j)} - z^{(j)} + \sum_{k=0}^{n-1} \sum_{\beta=0}^{1} (2\beta - 1) \frac{3 + (2\beta^{(k)} - 1)(2\beta - 1)}{2} \sigma(\tilde{w}^{(k)} x^{(j)} + \tilde{b}^{(k)})$$

$$= y^{(j)} - z^{(j)} + \sum_{k=0}^{n-1} (2\beta^{(k)} - 1) \sigma(\tilde{w}^{(k)} x^{(j)} + \tilde{b}^{(k)})$$

$$= y^{(j)} - z^{(j)} + g^{(n-1)}(x^{(j)})$$

$$= y^{(j)}.$$

In conclusion, we have therefore successfully identified weights and biases $(w, b) \in \mathcal{S}_X^A$ such that $F(w, b, v, X) = y$. $\square$

## D  DETAILS ON THE FUNCTION SPACE

We provide details on the discussion presented in Section 6.

*Proof of Proposition 13.* Consider first the case of a single ReLU. We write $x^{(i)}$ for the input data points in $1 \times \mathbb{R}$. The activation regions in parameter space are determined by the arrangement of hyperplanes $H_i = \{w \colon \langle w, x^{(i)} \rangle = 0\}$. Namely, the unit is active on the input data point $x^{(i)}$ if and only if the parameter is contained in the half-space $H_i^+ = \{w \colon \langle w, x^{(i)} \rangle > 0\}$ and it is inactive otherwise. We write $\bar{x}^{(i)} = [x_2^{(i)}, -1]$, which is a row vector that satisfies $\langle \bar{x}^{(i)}, x^{(i)} \rangle = 0$. We write $\mathbf{1}_S$ for a vector in $\mathbb{R}^{1 \times n}$ with ones at the coordinates $S$ and zeros elsewhere, and write $X_S = \mathbf{1}_S * X$ for the matrix in $\mathbb{R}^{d_0 \times n}$ where we substitute columns of $X$ whose index is not in $S$ by zero columns.

With these notations, in the following table we list, for each of the non-empty activation regions, the rank of the Jacobian, the activation pattern, the description of the activation region as an intersection of half-spaces, the extreme rays of the activation region, and the extreme rays of the function space represented by the activation region, which is simply the image of the Jacobian over the activation region.

| rank | $A$ | $\mathcal{S}_X^A$ | extreme rays of $\mathcal{S}_X^A$ | extreme rays of $\mathcal{F}_X^A$ | |
|---|---|---|---|---|---|
| 0 | $\mathbf{1}_\emptyset$ | $H_1^- \cap H_n^-$ | $\bar{x}^{(1)}, -\bar{x}^{(n)}$ | $0$ | |
| 1 | $\mathbf{1}_1$ | $H_1^+ \cap H_2^-$ | $\bar{x}^{(1)}, \bar{x}^{(2)}$ | $e_1$ | |
| 2 | $\mathbf{1}_{\leq i}$ | $H_i^+ \cap H_{i+1}^-$ | $\bar{x}^{(i)}, \bar{x}^{(i+1)}$ | $\bar{x}^{(i)} X_{\leq i-1}, \bar{x}^{(i+1)} X_{\leq i}$ | $(i = 2, \ldots, n-1)$ |
| 2 | $\mathbf{1}_{[n]}$ | $H_n^+ \cap H_1^+$ | $\bar{x}^{(n)}, -\bar{x}^{(1)}$ | $\bar{x}^{(n)} X_{\leq n-1}, -\bar{x}^{(1)} X_{\geq 2}$ | |
| 2 | $\mathbf{1}_{\geq i}$ | $H_{i-1}^- \cap H_i^+$ | $-\bar{x}^{(i)}, -\bar{x}^{(i-1)}$ | $-\bar{x}^{(i)} X_{\geq i+1}, -\bar{x}^{(i-1)} X_{\geq i}$ | $(i = 2, \ldots, n-1)$ |
| 1 | $\mathbf{1}_n$ | $H_{n-1}^- \cap H_n^+$ | $-\bar{x}^{(n)}, -\bar{x}^{(n-1)}$ | $e_n$ | |

The situation is illustrated in Figure 5 (see also Figure 1). On one of the parameter regions the unit is inactive on all data points so that the Jacobian has rank 0 and maps to 0. There are precisely two parameter regions where the unit is active on just one data point, $x^{(1)}$ or $x^{(n)}$, so that the Jacobian has rank 1 and maps to non-negative multiples of $e_1$ and $e_n$, respectively. On all the other parameter regions the unit is active at least on two data points. On those data points where the unit is active it can adjust the slope and intercept by local changes of the bias and weight and these are all the available degrees of freedom, so that the Jacobian has rank 2. These regions map to two-dimensional cones in function space. To obtain the extreme rays of these cones, we just evaluate the Jacobian on the two extreme rays of the activation region. This gives us item 1 in the proposition.

Consider now $d_1$ ReLUs. Recall that the Minkowski sum of two sets $M, N$ in a vector space is defined as $M + N = \{f + g \colon f \in M, g \in N\}$. An activation region $\S_X^A$ with activation pattern $A$

Figure 5: Subdivision of the parameter space of a single ReLU on two data points $x^{(1)}, x^{(2)}$ in $1 \times \mathbb{R}^1$ by values of the Jacobian (left) and corresponding pieces of the function space in $\mathbb{R}^2$ (right). The activation regions are intersections of half-spaces with activation patterns indicating the positive ones or, equivalently, the indices of data points where the unit is active.

for all units corresponds to picking one region with pattern $a^{(i)}$ for each of the units, $i = 1, \ldots, d_1$. Since the parametrization map is linear on the activation region, the overall computed function is simply the sum of the functions computed by each of the units,

$$F(W, v, X) = \sum_{i \in d_1} v_i F(w^{(i)}, X).$$

Here $F(W, v, X) = \sum_i v_i \sigma(WX)$ is the overall function and $F(w^{(i)}, X) = \sigma(w^{(i)}X)$ is the function computed by the $i$th unit. The parameters and activation regions of all units are independent of each other. Thus

$$\mathcal{F}_X^A = \sum_{i \in [d_1]} v_i \mathcal{F}_X^{a^{(i)}}.$$

Here we write $\mathcal{F}_X^{a^{(i)}} = \{(a_j^{(i)} w^{(i)} x^{(j)})_j \in \mathbb{R}^X : w^{(i)} \in \mathcal{S}_X^{a^{(i)}}\}$ for the function space of the $i$th unit over its activation region $\mathcal{S}_X^{a^{(i)}}$. This is a cone and thus it is closed under nonegative scaling,

$$\mathcal{F}_X^{a^{(i)}} = \alpha_i \mathcal{F}_X^{a^{(i)}} \quad \text{for all } \alpha_i \geq 0.$$

Thus, for an arbitrary linear combination of $d_1$ ReLUs we have

$$f = \sum_{i \in [d_1]} v_i f^{(i)} = \sum_{i \in [d_1]:\, f^{(i)} \neq 0} v_i \|f^{(i)}\|_1 \frac{f^{(i)}}{\|f^{(i)}\|_1}.$$

Here $f^{(i)}$ is an arbitrary function represented by the $i$th unit. We have $\sum_j f_j^{(i)} = \|f^{(i)}\|_1$ and $\sum_j f_j = \sum_i v_i \|f^{(i)}\|_1$. Thus if $f$ satisfies $f_1 + \cdots + f_n = 1$, then $\sum_i v_i \|f^{(i)}\| = 1$, and hence $f$ is an affine combination of the functions $\frac{f^{(i)}}{\|f^{(i)}\|}$. If all $v_i$ are non-negative, then $v_i \|f^{(i)}\| \geq 0$ and the affine combination is a convex combination. Each of the summands is an element of the function space of a single ReLU with entries adding to one.

In conclusion, the function space of a network with one hidden layer of $d_1$ ReLUs with non-negative output weights is the set of non-negative scalar multiples of functions in the convex hull of any $d_1$ functions in the normalized function space of a single ReLU. For a network with arbitrary output weights we obtain arbitrary scalar multiples of the affine hulls of any $d_1$ functions in the normalized function space of a single ReLU. This is what was claimed in items 2 and 3, respectively. □

# E    DETAILS ON THE EXPERIMENTS

We provide details on the experiments presented in Section 7. In addition, we provide experiments evaluating the number of activation regions that contain global optima, illustrating Theorem 12. Experiments were implemented in Python using PyTorch (Paszke et al., 2019), numpy (Harris et al., 2020), and mpi4py (Dalcin et al., 2011). The plots were created using Matplotlib (Hunter, 2007). The experiments in Section E.1 were run on the CPU of a MacBook Pro with an M2 chip and 8GB RAM. The experiments in Section E.2 were run on a CPU cluster that uses Intel Xeon IceLake-SP processors (Platinum 8360Y) with 72 cores per node and 256 GB RAM. The computer implementation of the scripts needed to reproduce our experiments can be found at https://anonymous.4open.science/r/loss-landscape-4271.

## E.1    NON-EMPTY ACTIVATION REGIONS WITH FULL RANK JACOBIAN

We sample a dataset $X \in \mathbb{R}^{d_0 \times n}$ whose entries are sampled iid Gaussian with mean 0 and variance 1. We use a two-layer network with weights $W \in \mathbb{R}^{d_1 \times d_0}$ and biases $b \in \mathbb{R}^{d_1}$ initialized iid uniformly on $\left[ -\frac{1}{\sqrt{d_1}}, \frac{1}{\sqrt{d_1}} \right]$. We choose a random activation region by evaluating $F$ at parameters $(W, b)$ and dataset $X$. Then we compute the Jacobian of $F$ on this activation region, and record whether or not it is of full rank. We repeat this 100 times to estimate the probability of the Jacobian being of full rank for a randomly sampled activation region. We calculate this probability for various values of $n$, $d_0$, and $d_1$. The results are reported in Figure 2 and Figure 3.

## E.2    NON-EMPTY ACTIVATION REGIONS WITH GLOBAL MINIMIZERS

We sample data $X, y$ as follows. The input data $X$ is generated as independent samples from a uniform distribution on the cube $[-1, 1]^{d_0}$. We consider three types of samples for the labels $y$:

- Polynomial: We construct a polynomial of degree 2 with the coefficients sampled from a uniform distribution on the interval $[-1, 1]$. The labels are then the evaluations of this polynomial at the points from $X$.
- Teacher: We construct a teacher network with an identical architecture to the main network used in the experiment. Then we initialize it with the same procedure as described below for the main network, which acts as a student network in this setup, and we take the outputs of the teacher network on $X$ as the labels.
- Random output: We sample labels from a uniform distribution on the interval $[-1, 1]$.

We sample activation regions as follows. For each experiment trial, we construct a ReLU fully-connected network with $d_0$ inputs, $d_1$ hidden units, and 1 output. Weights and biases of the hidden units are sampled iid from the uniform distribution on the interval $[-\sqrt{6/\text{fan-in}}, \sqrt{6/\text{fan-in}}]$ according to the uniform-He initialization (He et al., 2015). The weights of the output layer are initialized as alternating 1 and $-1$ and look like $[1, -1, 1, -1, \ldots]$. Additionally, we generate a new dataset $X, y$ for each experiment trial. Afterward, we consider an activation pattern $A$ corresponding to the network and the input data $X$.

For a given dataset $(X, y)$ and activation pattern $A$, we look for a zero-loss global minimizer to the linear regression problem: $\min_{\theta \in \mathbb{R}^{1 \times (d_0+1)d_1}} \frac{1}{2} \| \tilde{X}\theta - y^T \|_2^2$ subject to $\theta \in \mathcal{S}_X^A$, where $\theta \in \mathbb{R}^{(d_0+1)d_1 \times 1}$ is a flattened matrix of the first layer weights and $\tilde{X} \in \mathbb{R}^{n \times (d_0+1)d_1}$ is a vector with entries $\tilde{X}_{jk} = v_{k \bmod d_1} A_{j(k \bmod d_1)} X_{j \lfloor k/d_1 \rfloor}$, $y \in \mathbb{R}^{1 \times n}$. Here we appended the first layer biases to the weight rows and appended 1 to each network input. Following the descriptions given in Section 3, the second condition is a system of linear inequalities. Thus, this linear regression problem corresponds to the next quadratic program:

$$\min_{\theta \in \mathbb{R}^{1 \times (d_0+1)d_1}} \frac{1}{2} \theta^T P \theta + q^T \theta, \quad \text{where } P = \tilde{X}^T \tilde{X}, q = -\tilde{X}^T y^T$$

$$(2A_{ij} - 1)\langle w^{(i)}, x^{(j)} \rangle > 0 \quad \text{for all } i \in [d_1], j \in [n].$$

We record the percentage of sampled regions with a zero-loss minimizer. This is reported in Figures 6, 7, 8 for the three different types of data and $d_0 = 1, 2, 5$. The result is consistent with Theorem 12 in

that, for $d_0 = 1$, the minimal width $d_1$, needed for most regions to have a global minimum, is close to $d_1 \sim n \log n$ predicted by the theorem. We believe that this dependence will be relatively precise for most datasets. However, for specific data distributions, the probability of being in a region with a global minimum might be larger. In Figure 6, we see that, for instance, for data on a parabola, there are more interpolating activation regions than for data from a teacher network or random data. For higher input dimensions, we observe in Figures 7 and 8 that most of the regions contain a global minimum which is consistent with Theorem 5 and Corollary 8, by which any differentiable critical point in most non-empty activation regions is a global minimum.

Our figures are reminiscent of figures in the work of Oymak & Soltanolkotabi (2019). They showed for shallow ReLU networks that if the number of hidden units is large enough, $\sqrt{d_1 d_0} \geq C n^2 / d_0$, then gradient descent from random initialization converges to a global optimum. Empirically they observed a phase transition at roughly (but not exactly) $d_0 d_1 = n$ for the probability that gradient descent from a random initialization successfully fits $n$ random labels. Note that, in contrast, we are recording the number of regions that contain global optima.

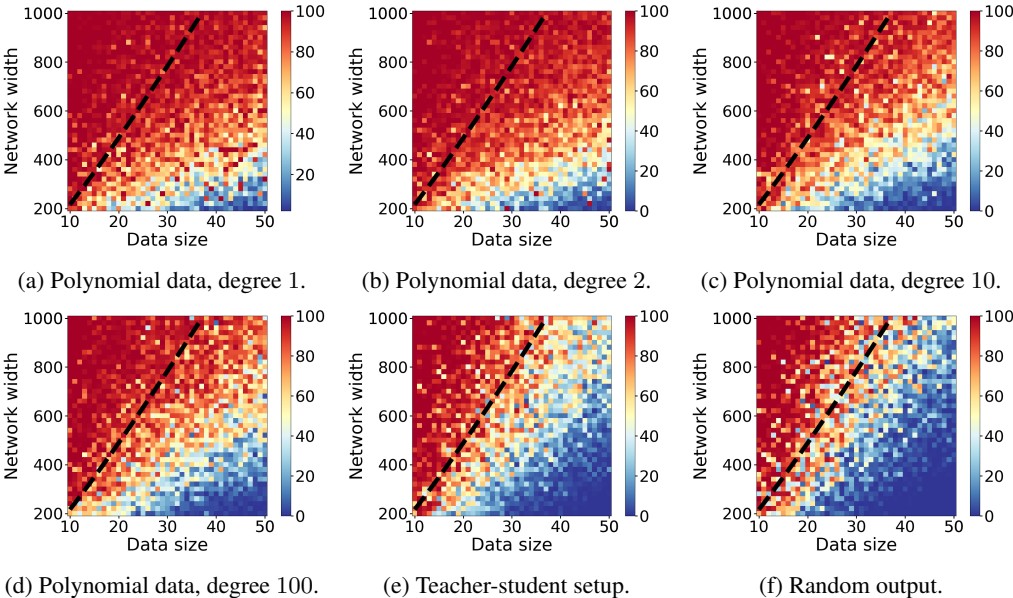

(a) Polynomial data, degree 1.  (b) Polynomial data, degree 2.  (c) Polynomial data, degree 10.

(d) Polynomial data, degree 100.  (e) Teacher-student setup.  (f) Random output.

Figure 6: Percentage of randomly sampled activation regions that contain a global minimum of the loss for the networks with input dimension $d_0 = 1$, depending on the size $n$ of the dataset and the width $d_1$ of the hidden layer. The results are based on $140$ random samples of the activation region for each fixed value of $n, d_1$. The target data is the same for each random network initialization for the same combination of $n$ and $d_1$. The black dashed line corresponds to the lower bound on $d_1$ estimated for a given $n$ and $\epsilon = 0.1$ based on the condition on the number of the negative and positive weights in the last network layer from Theorem 12. Precisely, it represents the function $d_1 = 4n \log(2n/\epsilon)$.

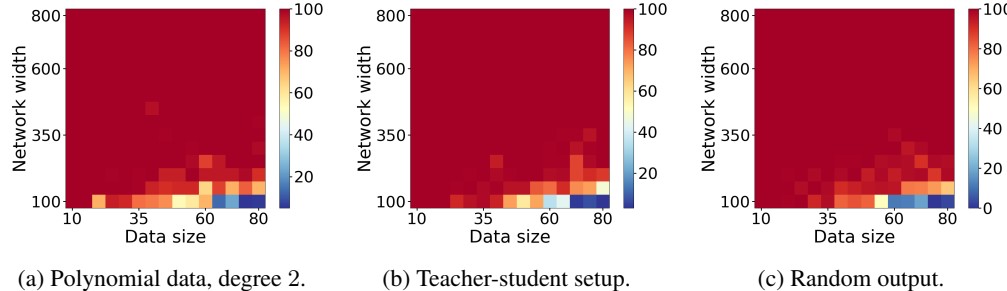

(a) Polynomial data, degree 2.    (b) Teacher-student setup.    (c) Random output.

Figure 7: Percentage of randomly sampled activation regions that contain a global minimum of the loss for the networks with input dimension $d_0 = 2$, depending on the size $n$ of the dataset and the width $d_1$ of the hidden layer. The results are based on $100$ random samples of the activation region for each fixed value of $n, d_1$. The target data is the same for each random network initialization for the same combination of $n$ and $d_1$.

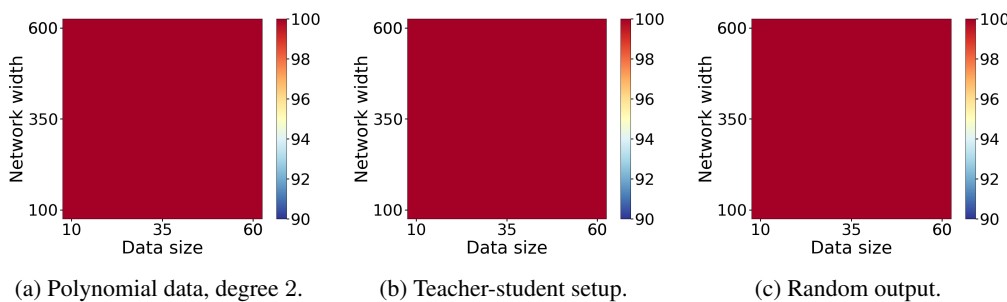

(a) Polynomial data, degree 2.    (b) Teacher-student setup.    (c) Random output.

Figure 8: Percentage of randomly sampled activation regions that contain a global minimum of the loss for the networks with input dimension $d_0 = 5$, depending on the size $n$ of the dataset and the width $d_1$ of the hidden layer. The results are based on $100$ random samples of the activation region for each fixed value of $n, d_1$. The target data is the same for each random network initialization for the same combination of $n$ and $d_1$.

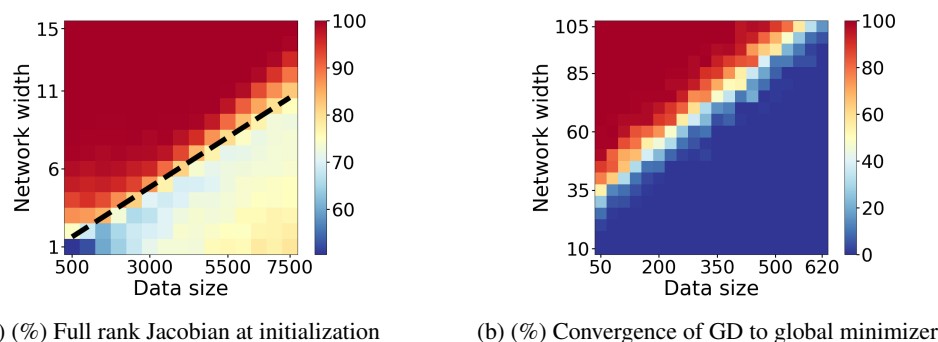

(a) (%) Full rank Jacobian at initialization    (b) (%) Convergence of GD to global minimizer

Figure 9: Classification task on MNIST to predict one hot binary class vectors. These heatmaps show percentages out of 100 trials of networks trained with GD from a random Gaussian initialization which have a full rank Jacobian at initialization (a) and achieve a cross-entropy loss of at most $10^{-2}$ after 1000 epochs (b). The number of network parameters matches the training set size $n$ when the width satisfies $d_1 = n/d_0$, where for MNIST the input dimension is $d_0 = 784$. As $d_0$ is large our results predict that there should exist some linear scaling of the network width $d_1$ and the data size $n$ such that in all but a very small fraction of regions every critical point is a global minimum.

# F   THE VOLUME OF ACTIVATION REGIONS

Our main results have concerned the number of activation regions containing bad local minima and the number of regions containing global minima. Here we will compute bounds for the volumes of these subsets of parameter space.

## F.1   ONE-DIMENSIONAL INPUT DATA

Consider the setting of Section 5, where we have a network $F(w, b, v, x)$ on one-dimensional input data. Recall that for $A \in \{0, 1\}^{d_1 \times n}$, we defined the activation region $\mathcal{S}_X^A$ by

$$\mathcal{S}_X^A := \{(w, b) \in \mathbb{R}^{d_1} \times \mathbb{R}^{d_1} : (2A_{ij} - 1)(w^{(i)}x^{(j)} + b^{(i)}) > 0 \text{ for all } i \in [d_1], j \in [n]\}.$$

For $k \in [n + 1]$, $\beta \in \{0, 1\}$ and corresponding step vectors $\xi^{(k,\beta)}$, we also define the individual neuron activation regions $\mathcal{N}_{k,\beta}$ by

$$\mathcal{N}_{k,\beta} := \{(w, b) \in \mathbb{R} \times \mathbb{R} : (2\xi^{(k,\beta)} - 1)(wx^{(j)} + b) > 0 \text{ for all } j \in [n]\}.$$

Let $\mu$ denote the Lebesgue measure. We bound the volumes of the activation regions intersected with the unit interval, that is, we bound the Lebesgue measures of the sets $\mathcal{S}_X^A \cap ([-1, 1]^{d_1} \times [-1, 1]^{d_1})$ and $\mathcal{N}_{k,\beta} \cap ([-1, 1] \times [-1, 1])$.

**Proposition 17.** *Let $\psi : [-\infty, \infty] \to [0, 2]$ be defined by*

$$\psi(x) := \begin{cases} -\frac{1}{2x} & \text{if } x \leq -1 \\ 1 + \frac{x}{2} & \text{if } -1 < x \leq 1 \\ 2 - \frac{1}{2x} & \text{if } x > 1 \end{cases}.$$

*Consider data $x^{(1)} \leq \cdots \leq x^{(n)}$. Then for $k \in [n + 1]$ and $\beta \in \{0, 1\}$,*

$$\mu(\mathcal{N}_{k,\beta} \cap ([-1, 1] \times [-1, 1])) = \psi(x^{(k)}) - \psi(x^{(k-1)}).$$

*Here we define $x^{(0)} := -\infty$, $x^{(n+1)} := \infty$.*

*Proof.* Let us define $\tilde{\psi} : [-\infty, \infty] \to [0, 2]$ by

$$\tilde{\psi}(x) := \mu\left(\left\{(w, b) \in (0, 1] \times [-1, 1] : \frac{b}{w} \leq x\right\}\right).$$

For $x \in \mathbb{R}$,

$$\tilde{\psi}(x) = \int_0^1 \int_{-1}^1 1_{b \leq wx} \, db \, dw$$

$$= \int_0^1 (1_{w \leq 1/|x|}(1 + wx) + 1_{wx \geq 1}(2)) \, dw$$

$$= \begin{cases} -\frac{1}{2x} & \text{if } x \leq -1 \\ 1 + \frac{x}{2} & \text{if } -1 < x \leq 1 \\ 2 - \frac{1}{2x} & \text{if } x \geq 1 \end{cases}$$

$$= \psi(x).$$

Moreover, $\tilde{\psi}(\infty) = 2 = \psi(\infty)$ and $\tilde{\psi}(-\infty) = 0 = \psi(\infty)$. So $\psi = \tilde{\psi}$. For all $k \in [n + 1]$, a neuron $(w, b)$ has activation pattern $\xi^{(k,0)}$ if and only if $wx^{(k-1)} + b > 0$ and $wx^{(k)} + b < 0$. So

$$\mu(\mathcal{N}_{k,0} \cap ([-1, 1] \times [-1, 1])) = \mu\left(\left\{(w, b) \in [-1, 1] \times [-1, 1] : wx^{(k-1)} + b < 0, wx^{(k)} + b > 0\right\}\right)$$

$$= \mu\left(\left\{(w, b) \in [0, 1] \times [-1, 1] : wx^{(k-1)} + b < 0, wx^{(k)} + b > 0\right\}\right)$$

$$= \mu\left(\left\{(w, b) \in [0, 1] \times [-1, 1] : wx^{(k-1)} - b < 0, wx^{(k)} - b > 0\right\}\right)$$

$$= \mu\left(\left\{(w, b) \in [0, 1] \times [-1, 1] : x^{(k-1)} < \frac{b}{w} < x^{(k)}\right\}\right)$$

$$= \psi(x^{(k)}) - \psi(x^{(k-1)}).$$

Similarly, a neuron $(w, b)$ has activation pattern $\xi^{(k,1)}$ if and only if $wx^{(k-1)} + b < 0$ and $wx^{(k)} + b > 0$. So

$$\mu(\mathcal{N}_{k,1} \cap ([-1, 1] \times [-1, 1])) = \mu\left(\left\{(w, b) \in [-1, 1] \times [-1, 1] : wx^{(k-1)} + b > 0, wx^{(k)} + b < 0\right\}\right)$$

$$= \mu\left(\left\{(w, b) \in [-1, 0] \times [-1, 1] : wx^{(k-1)} + b > 0, wx^{(k)} + b < 0\right\}\right)$$

$$= \mu\left(\left\{(w, b) \in [0, 1] \times [-1, 1] : -wx^{(k-1)} + b > 0, -wx^{(k)} + b < 0\right\}\right)$$

$$= \mu\left(\left\{(w, b) \in [0, 1] \times [-1, 1] : x^{(k-1)} < \frac{b}{w} < x^{(k)}\right\}\right)$$

$$= \psi(x^{(k)}) - \psi(x^{(k-1)}).$$

This establishes the result. $\qquad\square$

**Proposition 18.** *Let $n \geq 2$. Suppose that for all $j, k \in [n]$ with $j \neq k$, we have $|x^{(j)}| \leq 1$ and $|x^{(j)} - x^{(k)}| \geq \phi$. Then for all $k \in [n+1]$ and $\beta \in \{0, 1\}$,*

$$\mu(\mathcal{N}_{k,\beta} \cap ([-1, 1] \times [-1, 1])) \geq \frac{\phi}{4}.$$

*Moreover, for all $A \in \{0, 1\}^{d_1 \times n}$ whose rows are step vectors,*

$$\mu(\mathcal{S}_X^A \cap ([-1, 1]^{d_1} \times [-1, 1]^{d_1})) \geq \left(\frac{\phi}{4}\right)^{d_1}.$$

*Proof.* Since $n \geq 2$, $|x_1|, |x_2| \leq 1$, and $|x_1 - x_2| \geq \phi$, we have

$$\phi \leq |x_1 - x_2|$$
$$\leq |x_1| + |x_2|$$
$$\leq 2.$$

Let $\psi$, $x^{(0)}$, $x^{(n+1)}$ be defined as in Proposition 17. Fix $\beta \in \{0, 1\}$. If $k \in \{2, 3, \cdots, n\}$, then by Proposition 17 and the assumption that $|x^{(j)}| \leq 1$ for $j \in [n]$,

$$\mu(\mathcal{N}_{k,\beta} \cap ([-1, 1] \times [-1, 1])) = \psi(x^{(k)}) - \psi(x^{(k-1)})$$
$$= \frac{x^{(k)} - x^{(k-1)}}{2}$$
$$\geq \frac{\phi}{2}$$
$$\geq \frac{\phi}{4}.$$

If $k = 1$, then

$$\mu(\mathcal{N}_{k,\beta} \cap ([-1, 1] \times [-1, 1])) = \psi(x^{(1)}) - \psi(x^{(0)})$$
$$= 1 + \frac{x^{(1)}}{2}$$
$$\geq \frac{1}{2}$$
$$\geq \frac{\phi}{4}.$$

If $k = n + 1$, then

$$\mu(\mathcal{N}_{k,\beta} \cap ([-1, 1] \times [-1, 1])) = \psi(x^{(n+1)}) - \psi(x^{(n)})$$
$$= 1 - \frac{x^{(n)}}{2}$$
$$\geq \frac{1}{2}$$
$$\geq \frac{\phi}{4}.$$

Hence, for all $k \in [n+1]$ and $\beta \in \{0,1\}$, $\mu(\mathcal{N}_{k,\beta}) \geq \frac{\phi}{4}$.

The rows of $A$ are step vectors, so for each $i \in [d_1]$, there exists $k_i \in [n+1]$ and $\beta_i \in \{0,1\}$ such that
$$A_{i,\cdot} = \xi^{(k_i,\beta_i)}.$$

Then

$$\mu(\mathcal{S}_X^A \cap ([-1,1]^{d_1} \times [-1,1]^{d_1}))$$
$$= \mu(\{(w,b) \in [-1,1]^{d_1} \times [-1,1]^{d_1} : (2A_{ij}-1)(w^{(i)}x^{(j)} + b^{(i)}) > 0 \text{ for all } i \in [d_1], j \in [n]\})$$
$$= \prod_{i=1}^{d_1} \mu(\{(w,b) \in [-1,1] \times [-1,1] : (2A_{ij}-1)(wx^{(j)} + b) > 0 \text{ for all } j \in [n]\})$$
$$= \prod_{i=1}^{d_1} \mu(\{(w,b) \in [-1,1] \times [-1,1] : (2\xi_j^{(k_i,\beta_i)} - 1)(wx^{(j)} + b) > 0 \text{ for all } j \in [n]\})$$
$$= \prod_{i=1}^{d_1} \mu(\mathcal{N}_{k_i,\beta_i})$$
$$\geq \prod_{i=1}^{d_1} \frac{\phi}{4}$$
$$= \left(\frac{\phi}{4}\right)^{d_1}.$$

$\square$

The following proposition shows that in mildly overparameterized networks, a large subset of activation regions by volume have a full rank Jacobian.

**Proposition 19.** *Let $n \geq 2$. Suppose that the entries of $v$ are nonzero. Suppose that for all $j, k \in [n]$ with $j \neq k$, we have $|x^{(j)}| \leq 1$ and $|x^{(j)} - x^{(k)}| \geq \phi$. If*
$$d_1 \geq \frac{4}{\phi} \log\left(\frac{n}{\epsilon}\right),$$
*then*
$$\mu(\cup\{\mathcal{S}_X^A \cap [-1,1]^{d_1} \times [-1,1]^{d_1} : \nabla_{w,b} F \text{ has full rank on } \mathcal{S}_X^A\}) \geq (1-\epsilon)2^{2d_1}.$$

*Proof.* We use a probabilistic argument similar to the proof of Theorem 10. Let us choose a parameter initialization $(w,b) \in [-1,1]^{d_1} \times [-1,1]^{d_1}$ uniformly at random. Then the $i$-th row $A_{i,\cdot}$ of the activation matrix is a random step vector. By Proposition 18, for all $k \in [n+1]$ and $\beta \in \{0,1\}$,
$$\mathbb{P}(A_{i,\cdot} = \xi^{(k,\beta)}) \geq \frac{\phi}{4}. \tag{8}$$

By the proof of Theorem 10, the Jacobian of $\mathcal{S}_X^A$ is full rank when $A$ is a diverse matrix. That is, for all $k \in [n]$, there exists $i \in [d_1]$ such that $A_{i,\cdot} \in \{\xi^{(k,0)}, \xi^{(k,1)}\}$, and there exists $i \in [d_1]$ such that $A_{i,\cdot} = \xi^{(1,1)}$. For $i \in [d_1]$, let $C_i \in [n+1]$ be defined as follows. If $A_{i,\cdot} \in \{\xi^{(k,0)}, \xi^{(k,1)}\}$ for some $k \in \{2, 3, \cdots, n\}$, then we define $C_i = k$. If $A_{i,\cdot} = \xi^{(1,1)}$, then we define $C_i = 1$. Otherwise, we define $C_i = n+1$. By definition, $A$ is diverse if and only if
$$[n] \subseteq \{C_i : i \in [d_1]\}.$$

By (8), if $d_1 \geq \frac{4}{\phi} \log(\frac{n}{\epsilon})$, then for all $j \in [n]$,
$$\mathbb{P}(C_i = j) \geq \frac{\phi}{4}.$$

Thus, by Lemma 11,
$$\mathbb{P}(A \text{ is diverse}) = \mathbb{P}(n \subseteq \{C_i : i \in [d_1]\})$$
$$\geq 1 - \epsilon.$$

This holds for $(w, b)$ selected uniformly from $[-1, 1]^{d_1} \times [-1, 1]^{d_1}$. Hence, the volume of the union of regions with full rank Jacobian is at least

$$(1 - \epsilon)\mu([-1, 1]^{d_1} \times [-1, 1]^{d_1}) = (1 - \epsilon)2^{2d_1}.$$

$\square$

### F.2 ARBITRARY DIMENSION INPUT DATA

We now consider the setting of Section 3. Our model $(F \colon \mathbb{R}^{d_1 \times d_0} \times \mathbb{R}^{d_1}) \times \mathbb{R}^{d_0} \to \mathbb{R}$ is defined by

$$F(W, v, x) = v^T \sigma(Wx).$$

We consider the volume of the set of points $(W, v)$ such that $\nabla_{(W,v)} F(W, v, X)$ has rank $n$. We can formulate this problem probabilistically. Suppose that the entries of $W$ and $v$ are sampled from the standard normal distribution $\mathcal{N}(0, 1)$. We wish to compute the probability that the Jacobian has full rank. Our strategy will be to think of a random process which successively adds neurons to the network, and we will bound the amount of time necessary until the network has full rank with high probability.

**Definition 20.** For $\gamma \in (0, 1)$, we say that a distribution $\mathcal{D}$ on $\{0, 1\}^n$ is $\gamma$-*anticoncentrated* if for all nonzero $u \in \mathbb{R}^n$,

$$\mathbb{P}_{a \sim \mathcal{D}}(u^T a = 0) \leq 1 - \gamma.$$

**Lemma 21.** *Let $\gamma, \epsilon \in (0, 1)$. Suppose that $A \in \{0, 1\}^{d \times n}$ is a random matrix whose rows are selected iid from a distribution $\mathcal{D}$ on $\{0, 1\}^n$ which is $\gamma$-anticoncentrated. If*

$$d \geq \frac{8 \log(\epsilon^{-1})}{\gamma^2} + \frac{2n}{\gamma},$$

*then $A$ has rank $n$ with probability at least $1 - \epsilon$.*

*Proof.* Suppose that $a^{(1)}, a^{(2)}, \cdots \in \{0, 1\}^n$ are selected iid from $\mathcal{D}$. Define the filtration $(\mathcal{F}_t)_{t \in \mathbb{N}}$ by letting $\mathcal{F}_t$ be the $\sigma$-algebra generated by $a^{(1)}, \cdots, a^{(t)}$. For $t \in \mathbb{N}$, let $D_t$ denote the dimension of the vector space spanned by $a^{(1)}, \cdots, a^{(t)}$, and let

$$R_t := D_t - \gamma t.$$

Let $\omega \in \mathcal{F}_t$ be an event in which $a^{(1)}, \cdots, a^{(t)}$ do not span $\mathbb{R}^n$. Then there exists $u(\omega) \in \mathbb{R}^n$ nonzero such that $u^T a^{(s)} = 0$ for all $s \leq t$. Then

$$\begin{aligned} \mathbb{P}(D_{t+1} - D_t = 1 \mid \mathcal{F}_t)(\omega) &= \mathbb{P}(a^{(t+1)} \notin \text{span}(a^{(1)}, \cdots, a^{(t)}) \mid \mathcal{F}_t)(\omega) \\ &\geq \mathbb{P}(u^T a^{(t+1)} \neq 0 \mid \mathcal{F}_t)(\omega) \\ &= \mathbb{P}(u^T a^{(t+1)} \neq 0) \\ &\geq \gamma, \end{aligned}$$

where the third line from the independence of the $a^{(s)}$. Hence, for all $t \in \mathbb{N}$,

$$\begin{aligned} \mathbb{E}[1_{D_t \neq n}(D_{t+1} - D_t) \mid \mathcal{F}_t] &= 1_{D_t \neq n}\mathbb{E}[D_{t+1} - D_t \mid \mathcal{F}_t] \\ &= 1_{D_t \neq n}\mathbb{P}(D_{t+1} - D_t = 1 \mid \mathcal{F}_t) \qquad (9) \\ &\geq 1_{D_t \neq n}\gamma \end{aligned}$$

Let $\tau \in \mathbb{N}$ be a stopping time with respect to $(\mathcal{F}_t)_{t \in \mathbb{N}}$ defined by

$$\tau := \min(\{\infty\} \cup \{t \in \mathbb{N} : D_t = n\}).$$

We also define the sequence $(M_t)_{t\in\mathbb{N}}$ by $M_t := R_{\min(t,\tau)}$. Then for all $t \in \mathbb{N}$,

$$
\begin{aligned}
\mathbb{E}[M_{t+1} \mid \mathcal{F}_t] &= \mathbb{E}[R_{\min(t+1,\tau)} \mid \mathcal{F}_t] \\
&= \mathbb{E}[1_{\tau \leq t} R_\tau + 1_{\tau > t} R_{t+1} \mid \mathcal{F}_t] \\
&= 1_{\tau \leq t} R_\tau + \mathbb{E}[1_{\tau > t} R_{t+1} \mid \mathcal{F}_t] \\
&= 1_{\tau \leq t} R_\tau + \mathbb{E}[1_{D_t \neq n}(D_{t+1} - \gamma(t+1)) | \mathcal{F}_t] \\
&= 1_{\tau \leq t} R_\tau - 1_{D_t \neq n}\gamma(t+1) + \mathbb{E}[1_{D_t \neq n} D_{t+1} \mid \mathcal{F}_t] \\
&\geq 1_{\tau \leq t} R_\tau - 1_{D_t \neq n}\gamma(t+1) + 1_{D_t \neq n}\gamma + \mathbb{E}[1_{D_t \neq n} D_t \mid \mathcal{F}_t] \\
&= 1_{\tau \leq t} R_\tau - 1_{D_t \neq n}\gamma t + 1_{D_t \neq n} D_t \\
&= 1_{\tau \leq t} R_\tau + 1_{D_t \neq n} R_t \\
&= 1_{\tau \leq t} R_\tau + 1_{\tau > t} R_t \\
&= R_{\min(t,\tau)} \\
&= M_t,
\end{aligned}
$$

where we used (9) in the sixth line with properties of the conditional expectation. Moreover, we have $\mathbb{E}|M_t| < \infty$ for all $t \in \mathbb{N}$. Hence, the sequence $(M_t)_{t\in\mathbb{N}}$ is a submartingale with respect to the filtration $(\mathcal{F}_t)_{t\in\mathbb{N}}$. We also have $|M_{t+1} - M_t| \leq 1$ for all $t \in \mathbb{N}$. By Azuma-Hoeffding, for all $\delta > 0$ and $t \in \mathbb{N}$,

$$
\begin{aligned}
\mathbb{P}(M_t \leq -\delta) &\leq \mathbb{P}(M_t - M_1 \leq -\delta) \\
&\leq \exp\left(-\frac{\delta^2}{2(t-1)}\right) \\
&\leq \exp\left(-\frac{\delta^2}{2t}\right).
\end{aligned}
$$

So for any $\epsilon \in (0,1)$,

$$
\mathbb{P}(M_t \leq -\sqrt{2t\log(\epsilon^{-1})}) \leq \epsilon.
$$

We also have

$$
\begin{aligned}
\mathbb{P}(D_t \leq n-1) &= \mathbb{P}(D_t \leq n-1, \tau > t) \\
&= \mathbb{P}(R_t \leq n-1-\gamma t, \tau > t) \\
&= \mathbb{P}(M_t \leq n-1-\gamma t, \tau > t) \\
&\leq \mathbb{P}(M_t \leq n-1-\gamma t) \\
&\leq \mathbb{P}(M_t \leq n-\gamma t)
\end{aligned}
$$

So we have $D_t \leq n-1$ with probability at most $\epsilon$ when

$$
n - \gamma t \leq -\sqrt{2t\log(\epsilon^{-1})}.
$$

If

$$
t \geq \frac{8}{\gamma^2}\log(\epsilon^{-1}) + \frac{2n}{\gamma},
$$

then

$$
\begin{aligned}
\gamma t - \sqrt{2t\log(\epsilon^{-1})} &= \frac{\gamma}{2}t + \frac{\gamma\sqrt{t}}{2}\sqrt{t} - \sqrt{2t\log(\epsilon^{-1})} \\
&\geq \frac{\gamma}{2}\left(\frac{2n}{\gamma}\right) + \frac{\gamma\sqrt{t}}{2}\sqrt{\frac{8}{\gamma^2}\log(\epsilon^{-1})} - \sqrt{2t\log(\epsilon^{-1})} \\
&= n.
\end{aligned}
$$

Hence, for such values of $t$, we have $D_t \geq n$ with probability at least $1 - \epsilon$. In other words, with probability at least $1 - \epsilon$, the vectors $a^{(1)}, \cdots, a^{(t)}$ span $\mathbb{R}^n$. We can rephrase this as follows. If $A \in \{0,1\}^{d \times n}$ is a random matrix whose columns are selected iid from $\mathcal{D}$ and

$$
d \geq \frac{8}{\gamma^2}\log(\epsilon^{-1}) + \frac{2n}{\gamma},
$$

then with probability at least $1 - \epsilon$, $A$ has full rank. This proves the result. $\qquad\square$

Now we apply this lemma to study the rank of the Jacobian of the network. Let $\mathrm{sgn} : \mathbb{R} \to \{0, 1\}$ denote the step function:

$$\mathrm{sgn}(z) = \begin{cases} 0 & \text{if } z \leq 0 \\ 1 & \text{if } z > 0 \end{cases}.$$

For an input dataset $X \in \mathbb{R}^{d_0 \times n}$, consider the random variable

$$\mathrm{sgn}(X^T w),$$

where $w \sim \mathcal{N}(0, I)$, and $\mathrm{sgn}$ is defined entrywise. This defines the distribution of activation patterns on the dataset, which we denote by $\mathcal{D}_X$. We say that an input dataset $X \in \mathbb{R}^{d_0 \times n}$ is $\gamma$-anticoncentrated if for all nonzero $u \in \mathbb{R}^n$,

$$\mathbb{P}_{a \sim \mathcal{D}_X}(u^T a = 0) \leq 1 - \gamma.$$

We can interpret this as a condition on the amount of separation between data points. For example, suppose that two data points $x^{(j)}$ and $x^{(k)}$ are highly correlated: $\|x^{(j)}\| = \|x^{(k)}\| = 1$ and $\langle x^{(j)}, x^{(k)} \rangle \geq \rho$. Let us take $u \in \mathbb{R}^n$ to be defined by $u_i = 1$, $u_k = -1$, and $u_l = 0$ for $l \neq j, k$. Then

$$\begin{aligned} \mathbb{P}_{a \sim \mathcal{D}_X}(u^T a = 0) &= \mathbb{P}_{a \sim \mathcal{D}_X}(a_i = a_k) \\ &= \mathbb{P}_{w \sim \mathcal{N}(0, I)}(\mathrm{sgn}(w^T x^{(j)}) = \mathrm{sgn}(w^T x^{(k)})) \\ &= 1 - \frac{1}{\pi} \arccos(\langle x^{(j)}, x^{(k)} \rangle) \\ &\leq 1 - \frac{\arccos(\rho)}{\pi}. \end{aligned}$$

So in this case, the dataset is not $\gamma$-anticoncentrated for $\gamma = \frac{\arccos(\rho)}{\pi}$. In the other extreme, suppose that the dataset is uncorrelated: $\langle x^{(j)}, x^{(k)} \rangle = 0$ for all $j, k \in [n]$. Then for all $z \in \mathbb{R}$,

$$\begin{aligned} \mathbb{P}_{a \sim \mathcal{D}_X}(a_n = 0 \mid a_1, \cdots, a_{n-1}) &= \mathbb{P}_{w \sim \mathcal{N}(0, I)}(\langle w, x^{(n)} \rangle \leq 0 \mid \langle w, x^{(1)} \rangle, \cdots, \langle w, x^{(n-1)} \rangle) \\ &= \mathbb{P}_{w \sim \mathcal{N}(0, I)}(\langle w, x^{(n)} \rangle = 0) \\ &= \frac{1}{2}. \end{aligned}$$

For $u \in \mathbb{R}^n$ with $u_n \neq 0$,

$$\begin{aligned} \mathbb{P}_{a \sim \mathcal{D}_X}(u^T a = 0) &= \mathbb{E}[\mathbb{P}(u^T a = 0 \mid a_1, \cdots, a_{n-1})] \\ &= \mathbb{E}\left[ \mathbb{P}\left( u_n a_n = -\sum_{j=1}^{n-1} u_j a_j \middle| a_1, \cdots, a_{n-1} \right) \right] \\ &\leq \mathbb{E}[1/2] \\ &= 1/2. \end{aligned}$$

So in this case, the dataset is $\gamma$-anticoncentrated with $\gamma = 1/2$. In order to prove that the Jacobian has full rank with high probability, we must impose a separation condition such as this one – as data points get closer together, it becomes harder for the network to distinguish between them and the Jacobian drops rank. Once we impose $\gamma$-anticoncentration, a mildly overparameterized network will attain full rank at randomly selected parameters with high probability.

**Theorem 22.** *Let $\epsilon, \gamma \in (0, 1)$. Suppose that $X \in \mathbb{R}^{d_0 \times n}$ is generic and $\gamma$-anticoncentrated. If*

$$d_1 \geq \frac{8}{\gamma^2} \log\left(\frac{d_0}{\epsilon}\right) + \frac{2}{\gamma}\left(\frac{n}{d_0} + 1\right),$$

*then with probability at least $1 - \epsilon$, $\nabla_{(W, v)} F(W, v, X)$ has rank $n$.*

*Proof.* We employ the strategy used in the proof of Theorem 5. The Jacobian of $F$ with respect to $W$ is given by

$$\nabla_W F(W, v, X) = ((v \odot a^{(1)}) \otimes x^{(1)}, \cdots, (v \odot a^{(n)}) \otimes x^{(n)}),$$

where $a^{(j)}$ is the activation pattern of the $j$-th data point. With probability 1, none of the entries of $v$ are 0. So by Lemma 4, this Jacobian is of full rank when the set

$$\{a^{(i)} \otimes x^{(i)} : i \in [n]\}$$

consists of linearly independent elements of $\mathbb{R}^{d_1 \times d_0}$. We may partition $[n]$ into $d_0$ subsets $S_1, \cdots, S_{d_0}$ such that $|S_k| \leq \left\lceil \frac{n}{d_0} \right\rceil \leq \frac{n}{d_0} + 1$ for all $k \in [r]$ and partition the columns of $A$ accordingly into blocks $(a^{(s)})_{s \in S_k}$ for each $k \in [r]$. Let $M_k$ denote the $d_1 \times |S_k|$ matrix whose columns are the $a^{(s)}$, $s \in k$. Then the rows of $M_k$ are activation patterns of individual neurons of the network. So the rows of $M_k$ are iid and distributed according to $\mathcal{D}_X$. Since $\mathcal{D}_X$ is $\gamma$-anticoncentrated and

$$d_1 \geq \frac{8}{\gamma^2} \log \left( \frac{d_0}{\epsilon} \right) + \frac{2}{\gamma} \left( \frac{n}{d_0} + 1 \right)$$
$$\geq \frac{8}{\gamma^2} \log \left( \frac{d_0}{\epsilon} \right) + \frac{2|S_k|}{\gamma},$$

we have by Lemma 26 that with probability at least $1 - \frac{\epsilon}{d_0}$, $M_k$ has rank $|S_k|$. So with probability at least $1 - \epsilon$, all of the $M_k$ have full rank. This means that for each $k$,

$$\mathrm{rank}(\{a^{(i)} : i \in S_k\}) = |S_k|.$$

Then by Lemma 14,

$$\mathrm{rank}(\{a^{(i)} \otimes x^{(i)} : i \in [n]\}) = n,$$

so the Jacobian has rank $n$. $\qquad\square$

## G   NONSMOOTH CRITICAL POINTS

In the main body of this paper, our results considered the local minima in the interior of a given activation region. In this section we extend our analysis to handle points on the boundaries between regions where the loss is non-differentiable. We consider a network on univariate data $F(w, b, v, x)$ as in Section 5. Let $\mathrm{sgn} : \mathbb{R} \to \{0, 1\}$ be the unit step function:

$$\mathrm{sgn}(z) := \begin{cases} 0 & \text{if } z \leq 0 \\ 1 & \text{if } z > 0 \end{cases}.$$

Here we define the activation regions to include the boundaries:

$$\mathcal{S}_X^A := \left\{ (w, b) \in \mathbb{R}^{d_1} \times \mathbb{R}^{d_1} : \mathrm{sgn}(w^{(i)} x^{(j)} + b^{(i)}) = A_{ij} \text{ for all } i \in [d_1], j \in [n]. \right\}$$

For the rest of this section, we assume that the input dataset $X$ consists of distinct data points $x^{(1)} < x^{(2)} < \cdots < x^{(n)}$. By the same argument as Lemma 9, an activation region $\mathcal{S}_X^A$ is non-empty if and only if the rows of $A$ are step vectors.

**Theorem 23.** *Let $\epsilon \in (0, 1)$. If*

$$d_1 \geq 2n \log \left( \frac{n}{\epsilon} \right),$$

*then in all but at most a fraction $\epsilon$ of non-empty activation regions $A$, every local minimum of $L$ in $\mathcal{S}_X^A \times \mathbb{R}^{d_1}$ is a global minimum.*

*Proof.* We say that a binary matrix $A$ with step vector rows is *diverse* if for all $k \in [n]$, there exists $i \in [d_1]$ such that $A_{i, \cdot} = \xi^{(k, 1)}$. Suppose that $A$ is a binary matrix uniformly selected among all binary matrices with step vector rows. Define the random variables $C_1, \cdots, C_{d_1} \in [n + 1]$ by $C_i = k$ if $A_{i, \cdot} = \xi^{(k, 1)}$ for some $k \in [n]$, and $C_i = n + 1$ otherwise. Since the rows of $A$ are iid, the $C_i$ are iid. Moreover, for each $k \in [n]$, we have $\mathbb{P}(C_i = k) \geq \frac{1}{2n}$. So by Lemma 11, if

$$d_1 \geq 2n \log \left( \frac{n}{\epsilon} \right),$$

then with probability at least $1 - \epsilon$,

$$[n] \subseteq \{C_1, \cdots, C_{d_1}\}.$$

This means that for each $k \in [n]$, there exists $i$ such that $C_i = \xi^{(k,1)}$. In other words, $A$ is diverse. Since we chose $A$ uniformly at random among all binary matrices with step vector rows, it follows that all but a fraction $\epsilon$ of such matrices are diverse.

Now it suffices to show that when $A$ is diverse, every local minimum of $L$ in $\mathcal{S}_X^A \times \mathbb{R}^{d_1}$ is a global minimum. Note that $F$ is continuously differentiable with respect to $v$ everywhere. We will show that $\nabla_v F(w, b, v, X)$ has rank $n$ for all $(w, b, v) \in \mathcal{S}_X^A \times \mathbb{R}^{d_1}$. Since $A$ is diverse, there exist $i_1, \cdots, i_n \in [d_1]$ such that for all $k \in [n]$, $A_{i_k, \cdot} = \xi^{(k,1)}$. Consider the $n \times n$ submatrix $M$ of $\nabla_v F(w, b, v, X)$ generated by the rows $i_1, \cdots, i_n$. That is,

$$M_{pq} = \frac{\partial F}{\partial v^{(i_p)}}(w, b, v, x^{(q)}).$$

Then

$$M_{pq} = \sigma(w^{(i_p)} x^{(q)} + b^{(i_p)}),$$

so the entries of $M$ are non-negative, and

$$
\begin{aligned}
\operatorname{sgn}(M_{pq}) &= \operatorname{sgn}(w^{(i_p)} x^{(q)} + b^{(i_p)}) \\
&= A_{i_p, q} \\
&= (\xi^{(p,1)})_q \\
&= 1_{q \geq p}.
\end{aligned}
$$

Hence, $M$ is upper triangular with positive entries on its diagonals, implying that $\operatorname{rank}(M) = n$. Since $M$ is a submatrix of $\nabla_v F(w, b, v, X)$, we have $\operatorname{rank}(\nabla_v F(w, b, v, X)) = n$ as well. Now suppose that $(w, b, v) \in \mathcal{S}_X^A \times \mathbb{R}^{d_1}$ is a local minimum of $L$. Then

$$\nabla_v L(w, b, v, X) = \nabla_v F(w, b, v, X) \cdot (F(w, b, v, X) - y) = 0.$$

Since $\nabla_v F(w, b, v, X)$ has rank $n$, this implies that $F(w, b, v, X) = y$, and so $(w, b, v)$ is a global minimizer of $F$. So whenever $A$ is diverse, the region $\mathcal{S}_X^A \times \mathbb{R}^{d_1}$ has no bad local minima. This concludes the proof. □

## H   LOSS LANDSCAPES OF DEEP NETWORKS

While the results of this paper have focused on shallow networks, they admit generalizations to deep networks by considering the Jacobian with respect to individual layers of the network. In this section, we demonstrate this technique and prove that in deeper networks, most activation regions have no spurious critical points.

We consider fully connected deep networks with $L$ layers, where layer $l$ maps from $\mathbb{R}^{d_{l-1}}$ to $\mathbb{R}^{d_l}$, and $d_L = 1$. The parameter space consists of tuples of matrices

$$W = (W_1, W_2, \cdots, W_L)$$

where $W_l \in \mathbb{R}^{d_l \times d_{l-1}}$. We identify the vector space of all such tuples with $\mathbb{R}^m$, where $m = \sum_{l=1}^{L} d_l d_{l-1}$. For $l \in \{0, \cdots, L\}$, we define the $l$-th layer $f_l : \mathbb{R}^m \times \mathbb{R}^{d_{l-1}} \to \mathbb{R}^{d_l}$ recursively by

$$f_0(W, x) := x,$$
$$f_l(W, x) := \sigma(W_l f_{l-1}(\theta, x))$$

if $l \in [L-1]$, and

$$f_L(W, x) := v^T f_{l-1}(\theta, x),$$

where $v \in \mathbb{R}^{d_{l-1}}$ is a fixed vector whose entries are nonzero. Then $f_L$ is the final layer of the network, so the model is given by $F := f_L$. We denote the $j$-th row of $W_l$ by $w_l^{(j)}$. The activation patterns of a deep network are given by tuples

$$A = (A_1, A_2, \cdots, A_{L-1}),$$

where for each $l \in [L-1]$, $A_l \in \{0, 1\}^{d_l \times n}$. For an activation pattern $A$, let $\mathcal{S}_X^A$ denote the subset of parameter space corresponding to $A$. More precisely,

$$\mathcal{S}_X^A = \{W \in \mathbb{R}^m : (2A_{ij} - 1)\langle w_l^{(i)}, f_{l-1}(W, x^{(j)})\rangle > 0 \text{ for all } l \in [L-1], i \in [d_l], j \in [n]\}.$$

We say that a binary matrix $B$ is *non-repeating* if its columns are distinct.

**Proposition 24.** *Let $X \in \mathbb{R}^{d_0 \times n}$ be an input dataset with distinct points. Suppose that $A$ is an activation pattern such that $A_{L-1}$ has rank $n$, and such that $A_l$ is non-repeating for all $l \in [L-2]$. Then for all $W \in \mathcal{S}_X^A$, $\nabla_W F(W, X)$ has rank $n$.*

*Proof.* Suppose that $A$ is an activation pattern satisfying the stated properties. We claim that for all $W \in \mathcal{S}_X^A, l \in \{0, \cdots, L-2\}$, and $j, k \in [n]$ with $j \neq k$, that $f_l(W, x^{(j)}) \neq f_l(W, x^{(k)})$. We prove by induction on $l$. The base case $l = 0$ holds by assumption. Suppose that the claim holds for some $l \in \{0, \cdots, L-3\}$. By assumption, $A_{l+1}$ is non-repeating, so the columns $(A_{l+1})_{\cdot, j}$ and $(A_{l+1})_{\cdot, k}$ are not equal. Let $i \in [d_{l+1}]$ be such that $(A_{l+1})_{ij} \neq (A_{l+1})_{ik}$. Then, since $W \in \mathcal{S}_X^A$,

$$\text{sgn}(\langle w_{l+1}^{(i)}, f_l(W, x^{(j)})\rangle) \neq \text{sgn}(\langle w_{l+1}^{(i)}, f_l(W, x^{(k)})\rangle).$$

This implies that

$$\sigma(\langle w_{l+1}^{(i)}, f_l(W, x^{(j)})\rangle) \neq \sigma(\langle w_{l+1}^{(i)}, f_l(W, x^{(k)})\rangle),$$

or in other words

$$(f_{l+1}(W, x^{(j)}))_i \neq (f_{l+1}(W, x^{(k)}))_i.$$

So $f_{l+1}(W, x^{(j)}) \neq f_{l+1}(W, x^{(k)})$, proving the claim by induction.

Now we consider the gradient of $F$ with respect to the $(L-1)$-th layer. Let $\tilde{X} \in \mathbb{R}^{d_{L-2} \times n}$ be defined by $\tilde{X} := f_{L-2}(W, X)$, and for $j \in [n]$ let $\tilde{x}^{(j)}$ denote the $j$-th column of $\tilde{X}$. Let $a^{(1)}, \cdots, a^{(n)}$ denote the rows of $A_{L-1}$. Then for all $W \in \mathcal{S}_X^A$,

$$\nabla_{W_{L-1}} F(W, X) = ((v \odot a^{(1)}) \otimes \tilde{x}^{(1)}, \cdots, (v \odot a^{(n)}) \otimes \tilde{x}^{(n)}).$$

By Lemma 4, the rank of this matrix is equal to the rank of the matrix

$$(a^{(1)} \otimes \tilde{x}^{(1)}, \cdots, a^{(n)} \otimes \tilde{x}^{(n)}).$$

But $A_{L-1}$ has rank $n$ by assumption, so the set $a^{(1)}, \cdots, a^{(n)}$ is linearly independent, implying that the above matrix is full rank. Hence, $\nabla_{W_{L-1}} F(W, X)$ has full rank, and so $\nabla_W F(W, X)$ has full rank. $\square$

Now we count the number of activation patterns which satisfy the assumptions of Proposition 24 and hence correspond to regions with full rank Jacobian.

**Lemma 25.** *Suppose that $B \in \mathbb{R}^{d \times n}$ has entries chosen iid uniformly from $\{0, 1\}$. If*

$$d = \Omega\left(\log \frac{n}{\delta}\right),$$

*then with probability at least $1 - \delta$, $B$ is non-repeating.*

*Proof.* For any $j, k \in [n]$ with $j \neq k$,

$$\mathbb{P}(B_{\cdot, j} = B_{\cdot, k}) = \mathbb{P}(B_{ij} = B_{ik} \text{ for all } i \in [d])$$
$$= 2^{-d}.$$

So

$$\mathbb{P}(B \text{ is non-repeating}) = \mathbb{P}(B_{\cdot, j} \neq B_{\cdot, k} \text{ for all } j, k \in [n] \text{ with } j \neq k)$$
$$\geq 1 - \sum_{\substack{j, k \in [n] \\ j \neq k}} \mathbb{P}(B_{\cdot, j} = B_{\cdot, k})$$
$$\geq 1 - n^2 2^{-d}.$$

If

$$d \geq \frac{2 \log n + \log \frac{1}{\delta}}{\log 2},$$

then the above expression is at least $1 - \delta$. $\square$

**Lemma 26.** *Suppose that $B \in \mathbb{R}^{d \times n}$ has entries chosen iid uniformly from $\{0, 1\}$. If*

$$d = n + \Omega\left(\log \frac{1}{\delta}\right),$$

*then with probability at least $1 - \delta$, $B$ has rank $n$.*

*Proof.* Suppose that $d \geq n$. Let $B'$ be a $d \times d$ matrix selected uniformly at random from $\{0, 1\}^{d \times d}$, and let $B$ be the top $d \times n$ minor of $B'$. Note that $B$ has entries chosen iid uniformly from $\{0, 1\}$. Moreover, $B$ has rank $n$ whenever $B'$ is invertible. Moreover, by Theorem 3, $B'$ will be singular with probability at most $C(0.72)^d$, where $C \geq 1$ is a universal constant. Then

$$\mathbb{P}(\mathrm{rank}(B) = n) \geq \mathbb{P}(B' \text{ is invertible})$$
$$\geq 1 - C(0.72)^d.$$

Setting

$$d \geq n + \frac{\log \frac{C}{\delta}}{\log \frac{1}{0.72}},$$

we get that the above expression is at least

$$1 - C(0.72)^{\log(C/\delta)/\log(1/0.72)} = 1 - \delta.$$

Hence, if $d = n + \Omega(\log \frac{1}{\delta})$, then $B$ has rank $n$ with probability at least $1 - \delta$. $\qquad\square$

**Proposition 27.** *Let $X \in \mathbb{R}^{d_0 \times n}$ be an input dataset with distinct points. Suppose that for all $l \in [L - 2]$,*

$$d_l = \Omega\left(\log \frac{n}{\epsilon L}\right),$$

*and that*

$$d_{L-1} = n + \Omega\left(\log \frac{1}{\epsilon}\right).$$

*Then for at least a fraction $1 - \epsilon$ of all activation patterns $A$, the following holds. For all $W \in \mathcal{S}_X^A$, $\nabla_W F(W, X)$ has rank $n$.*

*Proof.* By Proposition 24, it suffices to count the fraction of activation patterns $A$ such that $A_l$ is non-repeating for $l \in [L - 2]$ and $A_{L-1}$ has rank $n$. Let $A$ be an activation pattern whose entries are chosen iid uniformly from $\{0, 1\}$. Fix $l \in [L - 2]$. Since $d_l = \Omega(\log(\frac{n}{\epsilon L}))$, by Lemma 25 we have with probability at least $1 - \frac{\epsilon}{2L}$ that $A_l$ is non-repeating. Hence, with probability at least $1 - \frac{\epsilon}{2}$, all of the $A_l$ for $l \in [L - 2]$ are non-repeating. Since $d_{L-1} = n + \Omega(\log \frac{1}{\epsilon})$, by Lemma 26 we have with probability at least $1 - \frac{\epsilon}{2}$ that $A_{L-1}$ has rank $n$. Putting everything together, we have with probability at least $1 - \epsilon$ that $A_l$ is non-repeating for $l \in [L - 2]$ and $A_{L-1}$ has rank $n$. So with this probability, $\nabla_W F(W, X)$ has rank $n$. Since we generated the activation pattern uniformly at random from all activation regions, the fraction of patterns $A$ with full rank Jacobian is at least $1 - \epsilon$.

$\qquad\square$

