# OpenReview forum: "Mildly Overparameterized ReLU Networks Have a Favorable Loss Landscape"
_ICLR.cc/2024/Conference — Submitted to ICLR 2024_

### Official Review · Reviewer_n6qD · 2023-10-25

**Soundness:** 3 good
**Presentation:** 3 good
**Contribution:** 3 good
**Rating:** 6
**Confidence:** 3

**Summary:**

This paper provides a landscape analysis for the differentiable regions of the two-layer ReLU network and proves the absence of bad local minima under mild overparameterization level.

**Strengths:**

1. The paper provides a novel perspective for the landscape analysis of 2-layer NN, from a geometric and combinatoric point of view. The theoretical analysis looks solid.

2. Figure 2 and 3 clearly show that the theorems are validated by numerical simulations.

**Weaknesses:**

1. The theoretical contribution can be better contextualized by readers if authors can provide some clarification on their intuitions. See bullet points 1 and 3 in Questions.

2. The key contribution, mild overparameterization level is only achieved for data of dimension 1. For the data with general dimension $d$ the counting method seems intractable. The authors are welcome to elaborate more on how to generate their analysis to high dimensional data or even deeper networks.

**Questions:**

1. I find it very difficult to follow the logic around Corollary 8: before Corollary 8, the authors claim that under general position assumption of the dataset and d larger than n, one can show that most activation regions are non-empty. However, Corollary 8 still focuses on the "non-empty activation regions". If most activation regions are indeed non-empty, why not drop the term "non-empty"?

2. Among the related works being provided in the first section, can you compare your work with Liu (2021)? They seem to provide a stronger result.

3. When $d_0=1$, Theorem 5 already provides some strong results, why do we bother deriving Theorem 10?

---

> ### Author Response · Authors · 2023-11-23
>
> Thanks for your helpful comments. We address all of your concerns and hope that our response clarifies your questions on our work.
>
> > The key contribution, mild overparameterization level is only achieved for data of dimension 1. For the data with general dimension the counting method seems intractable.
>
> We address the high-dimensional case in Theorem 5. One consequence of this result is that for high-dimensional data $(d_0 \geq n)$, if $d_1 \geq \Omega(1)$, then most activation regions have no bad critical points. This is a mild level of overparameterization since $d_0d_1 \sim n$.
>
> > The authors are welcome to elaborate more on how to generate their analysis to high dimensional data or even deeper networks.
>
> We have added a new Appendix H which generalizes our analysis to deep networks. Our methods for the shallow case can be applied to deeper networks by considering gradients with respect to individual layers. We find conditions on the activation patterns of each layer which ensure that the gradients will be full rank.
>
> > However, Corollary 8 still focuses on the ”non-empty activation regions”. If most activation regions are indeed non-empty, why not drop the term ”non-empty”?
>
> Indeed, if $d$ is large relative to $n$, then most activation regions will be non-empty. We include the term non-empty here to emphasize that our result is not just being applied to empty activation regions which trivially do not contain any critical points.
>
> > Among the related works being provided in the first section, can you compare your work with Liu (2021)? They seem to provide a stronger result.
>
> Liu (2021) shows that in a given activation region, all local minima are global *within the given region*. This means that for a critical point $\theta$ within an activation region $\Omega$, $\theta$ is a global minimizer of the loss restricted to $\Omega$, but there could still exist $\theta' \notin \Omega$ which attains a smaller loss than $\theta$. In contrast, we show that in most activation regions, all critical points are global minima *over the entire loss landscape*. We are proving a proving stronger statements about the critical points themselves by incorporating information about the activation patterns.
>
> > When $d_0 = 1$, Theorem 5 already provides some strong results, why do we bother deriving Theorem 10?
>
> In the one-dimensional case, most activation regions are empty (in contrast to the high-dimensional case where every activation pattern is realized). We provide a more refined analysis by focusing specifically on the non-empty activation regions, characterizing them, and showing that most of them have a full rank Jacobian.

---

> ### Comment · Reviewer_n6qD · 2023-12-02
>
> I thank the authors for providing the feedback. They have addressed my questions and I will keep my score.

---

### Official Review · Reviewer_fi53 · 2023-10-30

**Soundness:** 3 good
**Presentation:** 3 good
**Contribution:** 2 fair
**Rating:** 3
**Confidence:** 4

**Summary:**

This paper studies the loss landscape of the 1-hidden-layer ReLU network. The authors consider the partition of parameter space into different *activation regions*, defined by the activation patterns of ReLU neurons for different data points in the training dataset.

1. Under mild overparameterization, Theorem 5 in Section 3 shows that almost all activation regions have the property that every differentiable critical point of the training loss (with nonzero 2nd layer weights) is a global minimum; i.e., "no bad local minima" holds for most regions.

2. However, not all activation regions are non-empty (i.e., the inequalities defining the region are feasible). Section 4 studies the number of non-empty activation regions. For the high-dim case of $d \geq n$, for data points in general position, Theorem 5 can be extended to "almost all *non-empty* activation regions" (Corollary 8)

3. The paper then discusses the case of one-dimensional input + bias, in which stronger statements can be shown. The authors show in Theorem 10 that almost all non-empty activation regions satisfy the "no bad local minima" property. Under a stronger assumption on the 2nd layer weights, Theorem 12 shows existence of an affine set of global minima in almost all non-empty activation regions.

4. Some discussions on function space and experiments are provided in Sections 6 and 7, respectively.

**Strengths:**

S1. The paper shows that most linear regions in the parameter space satisfy the desirable "no bad local minima" property, which agrees with practical observations with overparameterized networks. The proof techniques (especially the one employing random sign matrices) seem to be new in the literature.

S2. The paper is relatively well-written and easy to digest.

**Weaknesses:**

W1. Unfortunately, the scope of this paper looks rather limited. The paper studies a single-hidden-layer model and only the first layer $W$ is trained. Hence, given an activation region, the prediction $F$ is linear in $W$, which makes the Jacobian matrix constant in the region and it makes proofs much easier. For Sections 5-6, the authors show results specialized to one-dimensional input space. I question if these proof techniques could be extended to deeper networks and higher-dimensional input space.

W2. The paper only considers differentiable local minima, which misses the possible existence of non-differentiable local minima. In fact, Laurent & von Brecht (2018) show that in the hinge loss case, local minima can only occur at the non-differentiable boundaries except for flat regions. Is there any hope of including non-differentiable points into the analysis?

W3. Most results in the paper only prove that for most activation regions, "all critical points are global minima". If we take a closer look at it, the theorems do not talk about the existence of critical points in these regions; they may or may not exist. Therefore, the statements cannot rule out the following pathological scenario: critical points do not exist at all in the $1-\epsilon$ fraction of the linear regions and the $\epsilon$ fraction of the regions contain bad local minima. In order to make the results stronger, a more complete characterization of the existence of critical points should be helpful.

W4. Some directly relevant citations are missing. [A, B, C] show the existence of bad local minima in ReLU networks and thus are directly relevant. [D] and some other papers by the same authors also consider the partition of parameter space with respect to the sign of pre-activations. I feel there should be more missing relevant papers, so please consider reviewing the literature again and updating the paper accordingly. Lastly, the citation to Safran & Shamir (2016) in page 1 should be corrected to Safran & Shamir (2018).

[A] Small nonlinearities in activation functions create bad local minima in neural networks

[B] Piecewise linear activations substantially shape the loss surfaces of neural networks

[C] Truth or Backpropaganda? An Empirical Investigation of Deep Learning Theory

[D] THE HIDDEN CONVEX OPTIMIZATION LANDSCAPE OF REGULARIZED TWO-LAYER RELU NETWORKS: AN EXACT CHARACTERIZATION OF OPTIMAL SOLUTIONS

**Questions:**

Please see the weaknesses above.

---

> ### Author Response · Authors · 2023-11-23
>
> We thank the reviewer for their valuable feedback and suggestions, which we address below. We hope that our response and updated manuscript address your concerns and you consider raising your score.
>
> > Unfortunately, the scope of this paper looks rather limited. The paper studies a single-hidden-layer model and only the first layer is trained. Hence, given an activation region, the prediction is linear in W, which makes the Jacobian matrix constant in the region and it makes proofs much easier. For Sections 5-6, the authors show results specialized to one-dimensional input space. I question if these proof techniques could be extended to deeper networks and higher-dimensional input space.
>
> > The paper only considers differentiable local minima, which misses the possible existence of non-differentiable local minima. In fact, Laurent \& von Brecht (2018) show that in the hinge loss case, local minima can only occur at the non-differentiable boundaries except for flat regions. Is there any hope of including non-differentiable points into the analysis?
>
> We have added appendices to address non-differentiable critical points and trainable output weights (Appendix G) as well as the deep case (Appendix H). Although our focus has been on the two-layer case, our techniques can be used to understand more general settings. Even when looking at non-differentiable points of activation regions, we can show in the case of one-dimensional input data that most activation regions contain no spurious minima.
>
> To generalize to the deep case, one can compute gradients with respect to individual layers of the network and find conditions on the activation patterns for which the gradients are full rank. We demonstrate this technique in Appendix H.
>
> We would also like to note that we consider a different setting from Laurent and von Brecht by considering the square loss rather than the hinge loss. When we restrict the hinge loss to a given activation region, the loss is locally given by an affine function, so there are no critical points on the interior of the region. In contrast, when using the square loss, the loss restricted to an activation region is given by a quadratic function, so a more careful analysis is needed.
>
> > Most results in the paper only prove that for most activation regions, "all critical points are global minima”. If we take a closer look at it, the theorems do not talk about the existence of critical points in these regions; they may or may not exist. Therefore, the statements cannot rule out the following pathological scenario: critical points do not exist at all in the fraction of the linear regions and the fraction of the regions contain bad local minima. In order to make the results stronger, a more complete characterization of the existence of critical points should be helpful.
>
> We address the issue of the existence of global minimizers in Theorem 12, where we show that with one-dimensional inputs, most activation regions contain global minimizers. In the one-dimensional case, we are able to show that $d_1 \sim n \log n$ is sufficient parameterization for most regions to have a full rank Jacobian and for the existence of global minima, with the existence of global minima requiring a constant factor of additional parameterization. For the higher-dimensional case, we provide experiments (in Appendix E.2) which confirm that approximately linear scaling of $d_1$ with $n$ is sufficient for most regions to contain global minima.
>
> > Some directly relevant citations are missing
>
> Thanks for highlighting these references. We have added pointers to these works in the introduction.

---

> > ### Comment · Reviewer_fi53 · 2023-12-01
> >
> > I appreciate the authors for their efforts in composing the rebuttal and also updating the paper; the addition of three appendices is a plus to the paper, and I think at least some of them should be incorporated into the main text to make the paper stronger.
> >
> > While the additional results partly resolve my concerns, there are still limitations to the scope of the paper, namely:
> > - Extension to deeper networks require that the last hidden layer is at least as wide as the number of data points $n$, which is a stronger requirement compared to the $n/d_0$ in Theorem 5.
> > - While I see the key differences in hinge loss in existing results, the paper's extension to non-differentiable regions as well as the existence proof of global minimizer are both limited to the 1-dimensional case. I believe this is a rather strong limitation and it seems highly nontrivial to extend the results on 1-dim input to higher dimensional cases.
> >
> > If we take out the limited $d_0 = 1$ case, the paper does not really say much about cases of general input dimension. Hence, I still feel reluctant to raise my score at this moment.

---

### Official Review · Reviewer_T8Cv · 2023-11-01

**Soundness:** 3 good
**Presentation:** 3 good
**Contribution:** 2 fair
**Rating:** 6
**Confidence:** 3

**Summary:**

This paper studies the loss landscape of two-layer ReLU networks. Following the observation that critical points where Jacobian is full rank are global minimizers, the authors show that most activation regions have no bad differentiable local minima. When the input is one-dimensional, they further proved that most regions contain a high-dimensional set of global minimizers. Experiments support that most regions of the loss landscape have full rank Jacobian in overparametrized networks.

**Strengths:**

-	Understanding the loss landscape of neural networks is a fundamental and important problem. This paper contributes a new angle by bounding the number of activation regions with no bad local minima.
-	The results are general as they are independent of specific choice of initialization of parameters or distribution of dataset.
-	The paper provides a rigorous definition for generic points using the fact that algebraic subset of $\mathbb{R}^n$ has measure 0. This method could be useful in various areas beyond studying landscape properties.

**Weaknesses:**

-	The setting described at the beginning of section 3 is different from a typical 2-layer ReLU network in that only one weight matrix is considered as parameters. It would be helpful to clarify whether including $v$ in the parameter space changes the conclusions in Theorem 5.
-	While smooth critical points are global minima in most of the activation regions, it is not clear what the volume of these regions are. In particular, results in this paper do not rule out the possibility that the small number of regions with bad local minima make up most of the parameter space. Quantifying the size of the activation regions appears difficult since the setting considered is independent of the distribution of dataset.
-	Despite interesting theoretical results, there is not much discussion on the implication on neural network training. As the authors also mention, this paper has not formalized how most regions having not bad local minima in their interior leads to gradient descent’s success in finding the global minima.

**Questions:**

-	The beginning of the second paragraph of section 6 states that a ReLU with one input with bias is equivalent to a ReLU with two input dimensions and no bias. Could the authors add a bit more explanation, as this equivalence does not seem straightforward?
-	Are the constants in the bound known? Can they be quantified in experiments?
-	This paper shows that most activation regions do not have bad local minima in their interior. Do there exist non-differentiable points that are local minimizers? If so, will these points cause problems for gradient descent?

---

> ### Author Response · Authors · 2023-11-23
>
> We thank the reviewer for their helpful comments which we address in our comments below and in the additions to our paper. In light of these improvements, we hope that you might consider increasing your score.
>
> > The setting described at the beginning of section 3 is different from a typical 2-layer ReLU network in that only one weight matrix is considered as parameters. It would be helpful to clarify whether including $v$
>  in the parameter space changes the conclusions in Theorem 5.
>
> We focused on computing the gradients with respect to the input weight matrix rather than the output head to facilitate our analysis. However, we would like to emphasize that our results showing that $\nabla_W F$ is full rank also imply that $\nabla_{(W, v)}F$ is full rank, since the former is a submatrix of the latter. In particular, our conclusions still hold when we include $v$. We have added a new Appendix G in which we analyze the effect of parameterizing $v$ as well as an extension to non-differentiable critical points.
>
> > While smooth critical points are global minima in most of the activation regions, it is not clear what the volume of these regions are. In particular, results in this paper do not rule out the possibility that the small number of regions with bad local minima make up most of the parameter space.
>
> Some of the methods we used to enumerate the activation regions with full rank Jacobian can be adapted to compute the volumes of such regions. We have added a new Appendix F illustrating this. For the one-dimensional case, we compute the volume of each activation region as a function of the dataset. We also bound the amount of parameterization needed for most of the parameter space (by volume) to have full rank Jacobian in terms of the separation between the data points. For the higher-dimensional case, we introduce an anticoncentration property of the data. Under this property, we prove that the portion of the loss landscape with full rank Jacobian is large by volume.
>
> > Despite interesting theoretical results, there is not much discussion on the implication on neural network training. As the authors also mention, this paper has not formalized how most regions having not bad local minima in their interior leads to gradient descent’s success in finding the global minima.
>
> From an optimization perspective, one significant benefit of most regions having full rank is that the neural tangent kernel in those regions is strictly positive definite. This means that while the parameters of the network remain in those regions, gradient flow will decay the square loss at an exponential rate.
>
> > The beginning of the second paragraph of section 6 states that a ReLU with one input with bias is equivalent to a ReLU with two input dimensions and no bias. Could the authors add a bit more explanation, as this equivalence does not seem straightforward?
>
> For a neural network on one-dimensional data with bias, each neuron is a function of the form $x \mapsto b + wx$. Instead, we could think of the data point $x$ as an element of 2-dimensional space, writing it as $(1, x)$. Then the mapping is of the form $(x_1, x_2) \mapsto bx_1 + wx_2 = (b, w)^T(x_1, x_2)$. The same trick applies in higher dimensions as well.
>
> ``Are the constants in the bound known? Can they be quantified in experiments?"
>
> For the one-dimensional case, we include the constants in our bounds for Theorems 10 and 12. For the high-dimensional case, we rely on a bound from Bourgain et al. [1] which states that the probability of a random $d \times d$ matrix being singular is at most $(1 / \sqrt{2} + o(1))^d$. Recovering the exact constant from this bound is challenging. However, the exact scaling can be checked experimentally, as we demonstrate in our experiments in Section 7. For example, Figure 3b shows that if we use the scaling $\frac{n}{d_0} = 2$, then $d_1 \approx 7$ is sufficient for at least 80 percent of activation regions to have no bad critical points. Note also that for fixed $\epsilon$, if $\frac{n}{d_0} \gg 1$, then Theorem 5 implies that $d_1 = \frac{n}{d_0}$ is sufficient for most regions to have a full rank Jacobian.
>
> > This paper shows that most activation regions do not have bad local minima in their interior. Do there exist non-differentiable points that are local minimizers? If so, will these points cause problems for gradient descent?
>
> Indeed, there can exist non-differentiable critical points which are local minimizers of the loss. We have added Appendix G to address this question, where we establish that in most activation regions, the set of spurious non-differentiable critical points forms a low-dimensional subset of the region.

---

> > ### Comment · Reviewer_T8Cv · 2023-12-02
> >
> > Thank you for the detailed response, which helped me better understand the paper’s contribution. I appreciate the added Appendix G that handles non-differentiable critical points. I am raising my score to 6.

---

### Official Review · Reviewer_Vqi8 · 2023-11-08

**Soundness:** 3 good
**Presentation:** 3 good
**Contribution:** 2 fair
**Rating:** 6
**Confidence:** 4

**Summary:**

This work studies the optimization landscape of mildly over-parameterized 2-layer ReLU networks. By looking at the rank of Jacobian in different activation regions, the authors claim that for most activation regions there are no bad differentiable local minima.

**Strengths:**

- The theoretical finding of this paper is rigorous and interesting. It provides good insight on the optimization landscape of mildly overparameterized NNs.
- The writing of this paper is easy to follow and the presentation is clear.

**Weaknesses:**

- The assumption of 2-layer network with fixed last layer weights $v$ is restrictive and impractical.
- The paper only discusses differentiable critical points, but there can be a lot of indifferentiable critical points depending on network settings. In fact generic deep ReLU networks with cross-entropy loss will have non-differentiable sub-optimal local minima [1]. This means that GD can still fall into bad non-differentiable local minima, even if the differentiable local minima are good. So it's still not clear to me that 'most of the landscape is favorable to optimization'.

[1] Bo Liu, Zhaoying Liu, Ting Zhang, Tongtong Yuan, Non-differentiable saddle points and sub-optimal local minima exist for deep ReLU networks, Neural Networks, Volume 144, 2021.

**Questions:**

- Continued on bad local minima. It seems to me that in figure 9(b) GD does not converge to good local minima when network is only mildly overparameterized, despite at initialization the Jacobian has full rank. Can you explain this? Does this contradict your theoretical findings?
- Numerical experiment.
-- How do you determine whether a Jacobian matrix is full rank?
-- For figure 9(a) is the network overparameterized at all? The experiment setting for figure 9 is not clear to me. Can you elaborate on this, just like what you did for other figures?

**Details Of Ethics Concerns:**

No ethics concerns.

---

> ### Author Response · Authors · 2023-11-23
>
> Thanks for your helpful comments and suggestions. We address each of your points below.
>
> > The assumption of 2-layer network with fixed last layer weights is restrictive and impractical.
> The paper only discusses differentiable critical points, but there can be a lot of indifferentiable critical points depending on network settings.
>
> We have added appendices to address non-differentiable critical points and trainable output weights (Appendix G) as well as the deep case (Appendix H). Although our focus has been on the two-layer case, our techniques can be used to understand more general settings. Even when looking at non-differentiable points of activation regions, we can show in the one-dimensional case that most regions have no spurious local minima. To generalize to the deep case, one can compute gradients with respect to individual layers of the network and find conditions on the activation patterns for which the gradients are full rank. We demonstrate this technique in Appendix H.  In Appendix F we now also discuss the volume of activation regions, providing further insight in regard to the comment "most of the landscape".
>
> > It seems to me that in figure 9(b) GD does not converge to good local minima when network is only mildly overparameterized, despite at initialization the Jacobian has full rank. Can you explain this? Does this contradict your theoretical findings?
>
> This experiment agrees with our theoretical results in the sense that when the network width $d_1$ scales linearly in the sample size $n$, gradient descent converges to a global minimizer with high probability. The amount of parameterization necessary for convergence to a global minimizer appears to be a constant multiple of the amount necessary for a full-rank Jacobian at initialization.
>
> > How do you determine whether a Jacobian matrix is full rank?
>
> We compute the Jacobian and its rank with PyTorch. The method ```torch.linalg.matrix_rank``` computes the SVD of the Jacobian and counts the number of singular values which are larger than a fixed machine tolerance.
>
> > For figure 9(a) is the network overparameterized at all? The experiment setting for figure 9 is not clear to me. Can you elaborate on this, just like what you did for other figures?
>
> Yes, the network is overparameterized in this experiment because the number of parameters is given by $m = d_0d_1$, where $d_0$ is the input dimension and $d_1$ is the hidden dimension. Here the dataset, MNIST, consists of images represented by 784-dimensional vectors. Hence the hidden dimension (network width) does not need to be large for the network to be overparameterized. We have updated the paper to clarify this by adding the comment "The number of network parameters matches the training set size $n$ when the width satisfies $d_1=n/d_0$, where for MNIST the input dimension is $d_0=784$."

---

### Author Response · Authors · 2023-11-23
**Common response: Summary of updates to the manuscript**

We would like to thank all of the reviewers for their helpful comments allowing us to improve our work. We have added three new appendices F, G, and H in the text addressing these issues and generalizing our existing results.

## Volumes of activation regions

The main results in our paper address the number of activation regions containing global minima and the number of regions with full rank Jacobian. A natural question to ask is whether these regions form sets of high volume, so that the network parameters are likely to be in one of the good regions at initialization. We answer this question in the affirmative in Appendix F. For one-dimensional data, we explicitly compute the volume of the activation regions, and derive bounds for the volume of good activation regions in terms of the amount of separation between data points. For multi-dimensional data, we introduce a condition on the input dataset under which most of the parameter space has full rank Jacobian with mild overparameterization. In both the one-dimensional case and the multi-dimensional case, our proofs for the volumes of activation regions closely mirror our proofs for the number of favorable activation regions, indicating that our techniques are general and not limited to only counting regions.

## Non-differentiable local minima

Our existing results show in multiple settings that most activation regions, every differentiable critical point is a global minimum. In Appendix G, in the case of one-dimensional data, we extend this analysis to the case of boundary points of regions while training both layers. As in the differentiable case, most regions have no spurious local minima.

## Loss landscapes of deep networks

We stated and proved our theorems for the case of two-layer ReLU networks. However, our techniques are not specific to this architecture and can be extended to the case of deep networks. In this setting, we compute gradients with respect to individual layers to find conditions on the activation patterns under which the Jacobian of the network is full rank. In Appendix H we find that when the second-to-last layer has width $\Omega(n)$ and the rest of the layers have width $\Omega(\log n)$, most activation regions have no bad local minima.

---

### Meta-Review · Area_Chair_3ZqQ · 2023-12-07

**Metareview:**

This study investigates the loss landscape of mildly overparameterized two-layer ReLU neural networks on finite input datasets using the squared error loss. The research employs techniques to bound the dimension of local and global minima sets based on the Jacobian rank. Results suggest that most activation patterns correspond to parameter regions without problematic differentiable local minima, especially for one-dimensional input data, where a significant portion of activation regions leads to high-dimensional global minima sets without bad local minima. Experimental findings support these results, revealing a phase transition influenced by the degree of overparameterization.


Four reviews were collected for this paper, and while most reviewers recognized the novelty of the main findings, significant concerns have been raised, which I summarize below:

1. Training on W but not v: Some reviewers noted that the paper's focus on the optimization landscape of 2-layer ReLU networks with a fixed vector v restricts the practical applicability of the results, as real networks are trained jointly on (W,v). Although the authors referred to Appendix G for an analysis of v parameterization and non-differentiable critical points, this section primarily addresses the latter and does not fully address the concern regarding varying v.

2. Focus on Differentiable Local Minima: Concerns were raised that the paper concentrates solely on differentiable local minima, overlooking the potential existence of non-differentiable ones. Given the prevalence of non-differentiable critical points in neural networks, this limitation narrows the paper's scope. While the authors attempted to address this in Appendix G, their focus on univariate data falls short of fully addressing this concern.

3. Volume of Activation Regions: Reviewers asked about the volume of activation regions without bad local minima and expressed uncertainty about whether the computed bounds in Appendix F have meaningful implications. A comprehensive discussion on these computed bounds remains lacking.

Addressing these concerns would necessitate a major revision of the paper, which could benefit from another comprehensive review. Therefore, I recommend rejection with the hope that the authors can address the comments raised and consider submitting the revised paper to another venue.

**Justification For Why Not Higher Score:**

There are several fundamental concerns about the paper that the authors should address prior to publications.

**Justification For Why Not Lower Score:**

N/A

---

### Decision · Program_Chairs · 2024-01-16

Reject